# Optimal Private and Communication Constraint Distributed Goodness-of-Fit Testing for Discrete Distributions in the Large Sample Regime

**Lasse Vuursteen**
Department of Statistics and Data Science
The Wharton School of the University of Pennsylvania
Philadelphia, PA 19104
`lassev@wharton.upenn.edu`

## Abstract

We study distributed goodness-of-fit testing for discrete distribution under bandwidth and differential privacy constraints. Information constraint distributed goodness-of-fit testing is a problem that has received considerable attention recently. The important case of discrete distributions is theoretically well understood in the classical case where all data is available in one "central" location. In a federated setting, however, data is distributed across multiple "locations" (e.g. servers) and cannot readily be shared due to e.g. bandwidth or privacy constraints that each server needs to satisfy. We show how recently derived results for goodness-of-fit testing for the mean of a multivariate Gaussian model extend to the discrete distributions, by leveraging Le Cam's theory of statistical equivalence. In doing so, we derive matching minimax upper- and lower-bounds for the goodness-of-fit testing for discrete distributions under bandwidth or privacy constraints in the regime where number of samples held locally are large.

**Keywords**: distributed inference, goodness-of-fit testing, differential privacy, communication constraint, federated learning, statistical equivalence.

## 1 Introduction

Federated learning is a fundamental problem in statistics and machine learning, where data is distributed across multiple locations (e.g. servers) and cannot readily be shared due to e.g. bandwidth or privacy constraints that each server needs to satisfy. The primary goal in these distributed data settings is to perform a single global inference task, such as hypothesis testing, regression, or classification, by aggregating the local information from each server.

Starting a few decades ago, investigations into distributed settings with bandwidth and other information constraints originated in the electrical engineering community, under the names "decentralized decision theory / the CEO problem" e.g. [90, 12, 92, 18, 58, 87] or "inference under multiterminal compression" (see [88] for an overview). These were largely motivated by applications where data is by construction observed and processed locally, such as astronomy, meteorology, seismology, surveillance systems, wireless communication, military radar or air traffic control systems.

Modern federated learning often involves data distributed across siloed data centers (e.g., hospitals) or networks of cellphone users, applied in areas such as word prediction, facial and voice recognition, virtual assistants like Siri or Google Assistant, autonomous vehicles, and earthquake prediction [65, 56, 51, 73, 67, 31]. In these settings, bandwidth often becomes a limited or costly resource [57].

Similarly, with advances in electronic record keeping, privacy has become a more and more pressing issue. These issues are prominent in tech industry products [32], including many federated learning

38th Conference on Neural Information Processing Systems (NeurIPS 2024).

applications mentioned earlier, as well as in scientific fields like medical sciences [60] and social sciences [69].

Methods that preserve privacy have been around in the statistics community for some time, starting in the 1980's [40, 41]. The current leading formal privacy framework is that of *differential privacy* (DP), as introduced in [42]. DP is a mathematical guarantee, describing whether results or data sets can be considered "privacy preserving" and hence can be openly published. Whilst many other privacy frameworks exist, this notion of privacy holds a prominent position both theoretically and practically, finding application within industry giants like Google [45], Microsoft [35], Apple [89], as well as governmental entities such as the US Census Bureau [74].

Quantifying the trade-off between privacy and statistical power means that researchers and data analysts can make an appropriate balance between data privacy and meaningful analysis. Similarly, by quantifying the impact of bandwidth constraints, systems can be designed to work as efficiently as possible within such bandwidth constraints.

The performance of distributed inference under bandwidth or differential privacy is well-studied for various estimation problems. For instance, distributed estimation under differential privacy has been studied for the many-normal-means model, discrete distributions and parametric models in [37, 38, 1, 98, 3], and density estimation [75, 59, 21], and nonparametric regression in [23]. Bandwidth constraints have been studied for the many-normal-means and parametric models in e.g. [101, 39, 77, 20, 97, 50, 26, 25], as well as nonparametric models, including Gaussian white noise [102], nonparametric regression [82], density estimation [17, 4], general, abstract settings [99] and online learning [95]. Distributed adaptive estimation methods under bandwidth constraints, where adaptation occurs to the unknown regularity of the functional parameter of interest, were derived in [82, 83, 25]. Testing simple hypotheses under bandwidth constraints has been studied by e.g. [92] and under differential privacy constraints by [29].

In this paper, we consider goodness-of-fit testing for discrete distributions (i.e. the multinomial model) in scenarios where the number of samples received by each server is large. Specifically, we study testing a simple null versus a composite alternative, in the setting where $m$ servers receive $n$ observations each from a distribution on a sample space of cardinality $d$, where $n$ is large comparatively to $m$ and $d$. Recently, such multinomial distributed data have found many applications in areas that handle very large samples over (possibly also large) discrete domains. For example, in population genetics [71, 86] and computer science; where it is used for e.g. information retrieval [100, 76], speech and text and classification [55], text mining [24] and large language models [72]. This has sparked recent interest in studying the statistical decision theoretic properties of the multinomial model, see [15] for an overview.

Deriving minimax rates for goodness-of-fit testing of discrete distributions under bandwidth and differential privacy constraints is particularly challenging when each server holds multiple observations. To date, matching rates have been established only when each machine observes a single observation [9, 10, 5] (see also our discussion of related work below). The techniques used to derive lower-bounds in the aforementioned paper heavily rely on the fact that each server contains only one observation, see [9]. Moreover, whilst tight lower-bounds for the multiple observations case exist for the Gaussian model, the functional analytic techniques used to derive these results heavily rely on Gaussianity, see [85] and [22] for the respective bandwidth constraint and DP lower-bounds. Additionally, lower-bound techniques developed for estimation problems generally do not yield tight impossibility results for goodness-of-fit testing problems (see also the discussion in Section B of the appendix).

We derive matching upper- and lower-bounds for goodness-of-fit testing for discrete distributions under bandwidth and differential privacy constraints in scenarios where the number of samples $n$ held by each of the servers is large in comparison to $d$ and $m$; $md \log d/\sqrt{n} = o(1)$. This is achieved by leveraging the theory of statistical equivalence, as introduced by Le Cam (see e.g. [62, 81] for an introduction). Leveraging existing results concerning statistical equivalence of multinomial data with a multivariate Gaussian model proven in [30] allow us to show, roughly speaking, that the distributed goodness-of-fit testing problem for discrete distributions is statistically equivalent a distributed goodness-of-fit testing problem for the mean of a multivariate Gaussian model, and hence the minimax rate for the former problem is the same as the minimax rate for the latter problem, which was established in [85] and [22], for bandwidth and differential privacy constraints respectively. Furthermore, we exploit the bandwidth constraint distributed setting in which these two models have

different minimax rates to show that, when $n$ is small compared to $d$ and $m$, the multinomial model and multivariate Gaussian model are statistically non-equivalent.

The rest of the paper is organized as follows. After a brief section on related work and notation, the article continues with a precise problem formulation in Section 2. Section 2.1 outlines the distributed framework, for both bandwidth and differential privacy constraints. In Section 2.2, we introduce the problem of distributed goodness-of-fit testing for discrete distributions under bandwidth and differential privacy constraints. In Section 3, the main results concerning the minimax rates are presented. Section 4 briefly outlines the main idea of the proof technique. Section 5 gives further insight into the comparison between discrete distributions and its comparable multivariate Gaussian model. The article ends with a brief discussion of the derived results. In the appendix, the tools of asymptotic equivalence within the distributed framework are presented and the technical proofs are provided.

## 1.1 Related work

Minimax goodness-of-fit testing knows a rich literature within the statistics and machine learning communities, see [46, 53, 64, 80, 52]. The $d$-ary discrete distribution uniformity testing problem bares a close relationship with "classical" nonparametric goodness-of-fit testing in the sense of [14, 79, 33, 96] and other nonparametric testing problems, see Section 1.4 in [54] and references therein.

For distributed goodness-of-fit testing specifically, much less is known. For multivariate Gaussian models under communication or differential privacy constraints, solutions have been established for the case where each server holds multiple observations. Communication constraints have been studied in [8, 84, 85] and differential privacy constraints in [22], with the authors deriving matching minimax upper- and lower-bounds for the goodness-of-fit testing for the mean of a multivariate Gaussian model.

For testing in discrete distributions, only the scenario where each server receives just one observation has been fully characterized in terms of the minimax rate in [9, 10, 5]. See also [28] for an overview. In these aforementioned works, the authors derive minimax rates goodness-of-fit testing for discrete distributions under bandwidth and differential privacy constraints. See [48, 78, 11, 2, 19] for investigations specifically under local DP (i.e. one observation per server with DP constraint). Nonparametric goodness-of-fit density testing for under local DP is considered in [36, 61], where in [61], the authors consider adaptation as well. For some investigations into the multiple observations per server case, see [34, 47].

For estimation, the bandwidth constraint estimation discrete distributions in the large sample-per-server case has been studied by [3], who derive matching upper and lower-bounds up to logarithmic factors. However, their technique does not extend to the goodness-of-fit testing problem.

## 1.2 Notation and notions

Throughout this paper, we shall use the following notation. For two positive sequences $a_k$ and $b_k$, we use $a_k \lesssim b_k$ to mean that $a_k \leq C b_k$ for some universal positive constant $C$. We write $a_k \asymp b_k$ if both $a_k \lesssim b_k$ and $b_k \lesssim a_k$, and $a_k \ll b_k$ if $a_k / b_k = o(1)$.

We denote the maximum of $a$ and $b$ by $a \vee b$ and the minimum by $a \wedge b$. For $k \in \mathbb{N}$, $[k]$ represents the set $\{1, \ldots, k\}$. Universal constants $c$ and $C$ may vary between lines. The Euclidean norm of a vector $v \in \mathbb{R}^d$ is denoted by $\|v\|_2$. For a matrix $M \in \mathbb{R}^{d \times d}$, $\|M\|$ represents the spectral norm, and $\mathrm{Tr}(M)$ denotes its trace. $I_d$ is the $d \times d$ identity matrix.

A non-negative sequence $M_k$ is said to be of poly-logarithmic order in non-negative sequences $a_k, b_k, c_k$ if there exists a constant $c > 0$ such that $M_k \lesssim (\log(a_k) \log(b_k) \log(c_k))^c$.

Given measurable spaces $(\mathcal{X}, \mathscr{X})$ and $(\mathcal{Y}, \mathscr{Y})$, a *Markov kernel $K$ (between $(\mathcal{X}, \mathscr{X})$ and target $(\mathcal{Y}, \mathscr{Y})$)* is a map $K \equiv K(\cdot|\cdot) : \mathscr{Y} \times \mathcal{X} \to [0, 1]$ with the following two properties: The map $x \mapsto K(A|x)$ is measurable for all $A \in \mathscr{Y}$, and the map $A \mapsto K(A|x)$ is a probability measure on $\mathscr{Y}$ for every $x \in \mathcal{X}$.

If $S$ is a random variable on a probability space $(\mathcal{X}, \mathscr{X}, \mathbb{P})$, we let $\mathbb{P}^S$ denote its *push-forward measure*, i.e. the measure defined by $\mathbb{P}^S(B) := \mathbb{P}(S^{-1}(B))$. We shall use $\mathbb{E}$ and $\mathbb{E}^S$ as the expectation

operator corresponding to $\mathbb{P}$ and $\mathbb{P}^S$. Random variables $X, Y, Z$ form a *Markov chain* $X \to Y \to Z$ whenever their joint distribution $\mathbb{P}^{(X,Y,Z)}$ disintegrates as $d\mathbb{P}^{(X,Y,Z)} = d\mathbb{P}^X d\mathbb{P}^{Y|X} d\mathbb{P}^{Z|Y}$.

## 2 Problem formulation

We begin by formally introducing the general framework of distributed inference.

### 2.1 The distributed framework

Consider a measurable space $(\mathcal{X}, \mathscr{X})$ with a statistical model $\mathcal{P} = \{P_f \ : \ f \in \mathcal{F}\}$ defined on it. In the distributed framework, we consider $j = 1, \ldots, m$ servers, each receiving data $X^{(j)}$ drawn from a given distribution $P_f \in \mathcal{P}$. Each of the servers communicates a transcript $Y^{(j)}$ based on the data to a central server, which in turn computes its solution to the testing problem $T(Y) \in \{0, 1\}$ based on the aggregated transcripts $Y = (Y^{(1)}, \ldots, Y^{(m)})$. We shall use the convention that $T(Y) = 1$ means rejecting the null hypothesis. The transcript generating mechanisms are then given by Markov kernels $\{K^j\}_{j=1,\ldots,m}$, with the Markov kernel (i.e. conditional distribution) of the transcript $Y^{(j)}$ given the data $X^{(j)}$ and the randomness $U$ shared by the servers denoted by $K^j(\cdot|X^{(j)}, U)$. We formalize this in the following definition.

**Definition 1.** *A* distributed testing protocol *for the model $\mathcal{P}$ consists of a triplet $\{T, \{K^j\}_{j=1}^m, (\mathcal{U}, \mathscr{U}, \mathbb{P}^U)\}$, where $\{K^j\}_{j=1}^m$ is a collection of Markov kernels $K^j : \mathcal{Y}^{(j)} \times \mathcal{X} \times \mathcal{U} \to [0, 1]$ defined on a measurable space $(\mathcal{Y}^{(j)}, \mathscr{Y}^{(j)})$, $T : \bigotimes_{j=1}^m \mathcal{Y}^{(j)} \to \{0, 1\}$ is a measurable map and $(\mathcal{U}, \mathscr{U}, \mathbb{P}^U)$ is probability space.*

The probability space $(\mathcal{U}, \mathscr{U}, \mathbb{P}^U)$ is used to (possibly) generate a source of randomness (independent of the data) that is shared by the servers. The distributed protocol is said to have *no access to shared randomness* or to be a *local randomness protocol* if $\mathbb{P}^U$ is trivial[1]. In an abuse of notation, we shall often refer to the entire triplet $\{T, \{K^j\}_{j=1,\ldots,m}, (\mathcal{U}, \mathscr{U}, \mathbb{P}^U)\}$ using just $T$.

Given a distributed protocol and i.i.d. data from $P_f$ we shall use $\mathbb{P}_f$ to denote the joint distribution of $Y = (Y^{(1)}, \ldots, Y^{(m)})$, the data $X$ under $P_f^m$ and the shared randomness $U \sim \mathbb{P}^U$. Writing $x = (x^{(1)}, \ldots, x^{(m)}) \in \mathcal{X}^m$, let $x \mapsto K(A|x, u)$ denote the Markov kernel $\bigotimes_{j=1}^m K^j(\cdot|x^{(j)}, u)$ (i.e. the product measure). The independence structure of the data yields that $P_f^m K = \bigotimes_{j=1}^m P_f K^j$ and the push-forward measure of $Y$ can be seen to disintegrate as

$$\mathbb{P}_f^Y(A) = P_f^m \mathbb{P}^U K(A) = \mathbb{P}^U P_f^m K(A) = \int \int K(A|x, u) dP_f^m(x) d\mathbb{P}^U(u),$$

where the second equality follows from the independence of $U$ with the data drawn from $P_f$. The above disintegration of the push-forward measure of $Y$ and the product structure of $K$ can be interpreted as $(X, Y, T(Y))$ forming a Markov chain given $U$, in the sense of the diagram

$$
\begin{array}{ccccc}
X^{(1)} & \longrightarrow & Y^{(1)}|U & \searrow & \\
\vdots & \longrightarrow & \vdots & \longrightarrow & T(Y). \\
X^{(m)} & \longrightarrow & Y^{(m)}|U & \nearrow &
\end{array}
\tag{1}
$$

The diagram indicates the flow of dependencies. The $m$ servers each obtain data $X^{(j)}$ from $P_f$, and generate a transcript $Y^{(j)}$ based on the data and shared randomness $U$. The central server then makes a decision $T(Y)$ based on the aggregated transcripts $Y$. For a definition of Markov kernels and Markov chains, see Section 1.2.

Allowing transcript-generating mechanisms to access both shared and local randomness is important for our analysis, as shared randomness has been found to yield strictly better performance in distributed goodness-of-fit testing, see e.g. [9, 10, 5, 8, 84, 85, 22]. Shared randomness protocols can be seen as a subset of common interactive procedures, such as sequential and blackboard protocols

---

[1] $\mathscr{U}$ is the trivial sigma-algebra (meaning $U \sim \mathbb{P}^U$ is a degenerate random variable)

(see e.g. [6]). The aforementioned paper shows that for discrete distribution goodness-of-fit testing in the single observation per server case, sequential and blackboard protocols offer no benefit over shared randomness protocols. Similarly, for mean shift problems in the multivariate Gaussian case, no advantage of sequential protocols over shared randomness protocols is known, except in the case of estimation with unknown variance [**?** ]. Since we study goodness-of-fit testing for discrete distributions in the large-number-of-observations case by comparing with a Gaussian model with known variance, we restrict the setting of the main article to local and shared randomness protocols only. Nevertheless, our theoretical framework is general enough to handle interactive protocols, which we discuss in Section A.2 of the appendix.

Next, we introduce the notion of a bandwidth constraint in the distributed setting.

**Definition 2.** *A distributed protocol is said to satisfy a $b$-bit bandwidth constraint if its kernels $\{K^j\}_{j=1,\dots,m}$ are defined on measurable spaces $(\mathcal{Y}^{(j)}, \mathscr{Y}^{(j)})$ satisfying $|\mathcal{Y}^{(j)}| \leq 2^b$ for $j = 1, \dots, m$.*

We use $\mathscr{T}_{\mathrm{LR}}^{(b)}$ and $\mathscr{T}_{\mathrm{SR}}^{(b)}$ to denote the classes of all local randomness and shared randomness distributed testing protocols with communication budget $b$ per machine, respectively.

Lastly, we introduce the notion of differential privacy in the distributed setting. We will be focusing on the notion of differential privacy as put forward by [43, 44]. Differential privacy provides a mathematical framework that guarantees preservation of privacy in a notion akin to cryptographical guarantees. Formally, a differential privacy constraint on a transcript in our setting is formulated as follows.

**Definition 3.** *Let $\epsilon \geq 0, \delta \geq 0$. The transcript $Y^{(j)}$ generated from $K^j, u \in \mathcal{U}$ is said to be $(\epsilon, \delta)$-differentially private if*

$$K^j(A|x, u) \leq e^\epsilon K^j(A|x', u) + \delta \tag{2}$$

$$\text{for all } A \in \mathscr{Y}^{(j)}, \ x, x' \in \mathcal{X}, \ i \in \{1, \dots, n\}.$$

*A distributed testing protocol $\{T, \{K^j\}_{j=1}^m, (\mathcal{U}, \mathscr{U}, \mathbb{P}^U)\}$, is said be a distributed $(\epsilon, \delta)$-differentially private testing protocol if $\{K^j\}_{j=1,\dots,m}$ satisfies (2) $\mathbb{P}^U$-a.s.*

Small values of $\epsilon$ and $\delta$ ensure that, even when the transcript $Y^{(j)}$ is publicly available, the sample $X^{(j)}$ underlying $Y^{(j)}$ is unidentifiable. We stress that this type of differential privacy guarantee concerns the local data $X^{(j)}$ in full, even $X^{(j)}$ consists of multiple observations. This is often referred *local differential privacy*, where the privacy guarantee regards each server as essentially pertaining data to "one indiviual". For a thorough introduction on differential privacy guarantees, we refer the reader to [42]. We also note that the use of shared randomness does not affect the privacy guarantee provided by the protocol, as the guarantee holds even if the outcome of the shared randomness is known.

We use $\mathscr{T}_{\mathrm{LR}}^{(\epsilon, \delta)}$ and $\mathscr{T}_{\mathrm{SR}}^{(\epsilon, \delta)}$ to denote the classes of all local- and shared randomness $(\epsilon, \delta)$-differentially private distributed testing protocols, respectively. We note that the machinery developed in Section A.2 allows consideration of both types of constraints simultaneously. In the main text of the article, we shall focus on the bandwidth constraint and differential privacy constraint separately as minimax rates for the joint constraints are not known for the Gaussian model we use for comparison to the multinomial model in the main article.

## 2.2 Distributed goodness-of-fit testing

We start by giving a formal description of sampling from a discrete distribution in the distributed setting. Consider a set with cardinality $d$; for simplicity, we take $\tilde{\mathcal{X}} = \{1, \dots, d\}$. Any probability distribution such a set can be characterized by an element of the $d-1$-dimensional probability simplex $\mathbb{S}^d$, defined as

$$\left\{ q = (q_1, \dots, q_d) \in [0, 1]^d : \sum_{i=1}^d q_i = 1 \right\}.$$

In our distributed framework, each server $j = 1, \dots, m$ observes a data $\tilde{X}^{(j)}$ taking values in $\{1, \dots, d\}^n$

$$\tilde{X}^{(j)} = (\tilde{X}_1^{(j)}, \dots, \tilde{X}_n^{(j)}) \sim Q \equiv Q_{n,q}, \quad \tilde{X}_i^{(j)} \overset{i.i.d.}{\sim} \text{Multinomial}(1, q) \ \text{for } q \in \mathbb{S}^d. \tag{3}$$

That is, each server obtains $n$ i.i.d. draws from a multinomial distribution with parameter $q$.

The statistical decision problem of interest shall be that of *goodness-of-fit* or *uniformity testing*, i.e. distinguishing the hypotheses

$$H_0 : q = q_0 \text{ versus } H_1 : q \in \{q \in \mathcal{F} : \|q - q_0\|_1 \geq \rho\} =: H_\rho, \tag{4}$$

where $q_0 = (q_{01}, \ldots, q_{0d}) = (1/d, \ldots, 1/d) \in \mathbb{S}^d$ and

$$\mathcal{F} = \left\{ q \in \mathbb{S}^d \ : \ \frac{\max_i q_i}{\min_i q_i} \leq R \right\}, \tag{5}$$

for some fixed constant $R > 0$. The statistical model under consideration shall be denoted by $\mathcal{Q} = \{Q_q^n : q \in \mathcal{F}\}$.

We define the testing risk for a distributed testing protocol $T$, for the hypotheses (4) (and statistical model $\mathcal{Q}$) by sum of the type I and worst case type II error over the alternative class;

$$\mathcal{R}_{\mathcal{Q}}(T, H_\rho) := \mathbb{Q}_{q_0}^Y T(Y) + \sup_{f \in H_\rho} \mathbb{Q}_f^Y \left(1 - T(Y)\right).$$

The minimax testing risk over a class of distributed protocols $\mathcal{T}$ is then defined as $\inf_{T \in \mathcal{T}} \mathcal{R}_{\mathcal{Q}}(T, H_\rho)$.

It is clear that, as $\rho$ tends to 0, the minimax testing risk should increase. We are interested in finding the so called *minimax separation rate*, or detection boundary, which is a sequence $\rho^*$ depending on the model characteristics $n, d, m$ and $\mathcal{T}$ such that the minimax testing risk converges to 0 if $\rho \ll \rho^*$ or 1 if $\rho \gg \rho^*$.

The minimax separation rate captures how the testing problem becomes easier, or more difficult, for different model characteristics. The minimax rate for the hypothesis above case is $\rho^2 \asymp \frac{\sqrt{d}}{mn}$ when $\mathcal{T}$ consists of the class of all testing protocols, as was established in [70] and [94].

When $\mathcal{T}$ is taken to be one of the bandwidth or privacy constraint classes of tests, i.e. $\mathcal{T}_{\text{LR}}^{(b)}$ and $\mathcal{T}_{\text{SR}}^{(b)}$ $\mathcal{T}_{\text{LR}}^{(\epsilon,\delta)}$ and $\mathcal{T}_{\text{SR}}^{(\epsilon,\delta)}$, it is sensible to expect $\rho^*$ to depend on the bandwidth or differential privacy parameters, $b$ and $(\epsilon, \delta)$, respectively. In the distributed discrete distribution setupj described above with $n = 1$, such minimax rates have been derived in [9, 10]. We discuss these results in the next section, contrasting them with the minimax separation rate derived in this paper for the case where $md \log d / \sqrt{n} = o(1)$.

## 3 Minimax rates in the large sample regime

We now turn to the main results of this paper, which concern the minimax rates for goodness-of-fit testing for discrete distributions under bandwidth and differential privacy constraints in the large sample regime. We shall show that the minimax rates for the distributed multinomial model under bandwidth and differential privacy constraints are the same as the minimax rates for a $d$-dimensional distributed Gaussian model, as derived in [85] and [22], respectively.

The first theorem establishes the minimax rate for the distributed multinomial model under bandwidth constraints. A proof can be found in Section D of the appendix.

**Theorem 1.** *Consider sequences $m \equiv m_\nu$, $b \equiv b_\nu$, $d \equiv d_\nu$ and $n \equiv n_\nu$ such that $md \to \infty$ whilst*

$$md \log d / \sqrt{n} \stackrel{\nu \to \infty}{\to} 0.$$

*Suppose that $\rho \equiv \rho_\nu$ is a nonnegative sequence satisfying*

$$\rho^2 \asymp \left( \frac{d}{\sqrt{d \wedge b}mn} \right) \bigwedge \left( \frac{\sqrt{d}}{\sqrt{m}n} \right). \tag{6}$$

*Then,*

$$\inf_{T \in \mathcal{T}_{SR}^{(b)}} \mathcal{R}_{\mathcal{Q}}(T, H_{M_\nu \rho}) \to \begin{cases} 0 \ \text{for any } M_\nu \to \infty, \\ 1 \ \text{for any } M_\nu \to 0. \end{cases}$$

*When considering the class of only local randomness protocols (i.e. replacing $\mathscr{T}_{SR}^{(b)}$ with $\mathscr{T}_{LR}^{(b)}$ in the above display), the minimax separation rate is given by*

$$\rho^2 \asymp \left( \frac{d^{3/2}}{(d \wedge b)mn} \right) \bigwedge \left( \frac{\sqrt{d}}{\sqrt{mn}} \right). \tag{7}$$

The theorem above shows that the minimax rate for the distributed multinomial model under bandwidth constraints is given by (6) in the case of access to shared randomness, and (7) in the case of no access to shared randomness. Both rates are the same as those established for a signal detection problem in a $d$-dimensional distributed Gaussian model, as derived in [85], Theorems 3.1 and 3.2. In Section 4, we shall provide a proof of this result through the notion of statistical equivalence, where we explicitly use that the multinomial model is asymptotically similar to a specific Gaussian model and a corresponding signal detection problem.

The distributed $b$-bit bandwidth constraint minimax rate for the hypotheses (4) in the multinomial model with $n = 1$ is established in [9, 10]. Specifically, they find that

$$\rho^2 \asymp \begin{cases} \frac{d}{m\sqrt{2^b \wedge d}} & \text{in case of access to shared randomness,} \\ \frac{d\sqrt{d}}{m(2^b \wedge d)} & \text{without access to shared randomness.} \end{cases} \tag{8}$$

Several aspects of this minimax rate are intriguing. First, unlike in the "large $n$ case" for the same model and hypothesis ((6) and (7)), there is no elbow effect. Secondly, the benefit (i.e. "efficiency gain") from an increase in bandwidth is exponential, whereas in the large sample scenario of Theorem 4 it is sub-linear. We shall comment on this "communication super-efficiency" phenomenon further below.

We now turn to the distributed multinomial model under differential privacy constraints. As in the case of the bandwidth constraint uniformity testing problem, we shall show that the minimax rate for the distributed multinomial model under differential privacy constraints is the same as the minimax rate for a $d$-dimensional distributed Gaussian model, as derived in [22].

The following theorem describes that the above rates are the minimax rates for uniformity testing in the distributed multinomial model under differential privacy constraints, for shared randomness and local randomness only protocols, respectively.

**Theorem 2.** *For any sequences $m \equiv m_\nu$, $d \equiv d_\nu$ and $n \equiv n_\nu$ such that $md \to \infty$, $\frac{md \log d}{\sqrt{n}} \overset{\nu \to \infty}{\to} 0$, $n^{-1/4} \ll \epsilon \equiv \epsilon_\nu \leq 1$, $\delta \equiv \delta_\nu \asymp (md)^{-p}$ for some $p > 1$. The minimax separation rate in the distributed multinomial model $\mathcal{Q}$ for testing the hypotheses (4) using locally $(\epsilon, \delta)$-differentially private protocols is*

$$\rho^2 \asymp poly\text{-}log(d, m, n) \begin{cases} \frac{d}{mn\epsilon^2} & \text{if } \epsilon \geq \frac{\sqrt{d}}{\sqrt{m}}, \\ \frac{\sqrt{d}}{\sqrt{mn}\epsilon} & \text{if } \frac{1}{\sqrt{md}} \leq \epsilon < \frac{\sqrt{d}}{\sqrt{m}} \end{cases} \tag{9}$$

*in the case of having access to shared randomness. In the case of having only access to local randomness, it is given by*

$$\rho^2 \asymp poly\text{-}log(d, m, n) \begin{cases} \frac{d\sqrt{d}}{mn\epsilon^2} & \text{if } \epsilon \geq \frac{d}{\sqrt{m}}, \\ \frac{\sqrt{d}}{\sqrt{mn}\epsilon} & \text{if } \frac{1}{\sqrt{md}} \leq \epsilon < \frac{d}{\sqrt{m}}. \end{cases} \tag{10}$$

We provide a proof of the theorem in Section D of the appendix. As with the bandwidth constraint case, the minimax separation rates for the distributed multinomial model under differential privacy constraints are derived by comparing the model and hypothesis test to a signal detection problem for the $d$-dimensional distributed Gaussian model. The rates for the latter problem follow from the proofs of Theorems 4 and 5 in [22], who describe a more general setup which includes signal detection in the $d$-dimensional distributed Gaussian model as a special case[2].

Also in the case of privacy, there is a difference between the one observation per server case minimax rate ($n = 1$) and the multiple observations per server with local differential privacy case. The

---

[2]The above rates do not observe all phase transitions present in the more general setup of [22], as it pertains to the local differential privacy case in that paper, with $\sigma = 1/\sqrt{n}$ in their notation.

minimax rate in the multinomial model for $n = 1$ is derived in [9, 5];

$$\rho^2 \asymp \begin{cases} \frac{d}{m\epsilon^2} & \text{in case of access to shared randomness,} \\ \frac{d\sqrt{d}}{m\epsilon^2} & \text{without access to shared randomness.} \end{cases} \tag{11}$$

Comparing this rate to the rate obtained in Theorem 2, we observe phase transitions in the distributed testing problem for multinomial model under local differential privacy constraints which only occurs if the number of observations locally is large compared to the cardinality of the sample space.

## 4 Deriving the minimax rates through statistical equivalence

The minimax rates for the distributed multinomial model under bandwidth and differential privacy constraints are derived through the notion of statistical equivalence (Le Cam theory), which is a powerful tool for establishing minimax rates in statistical decision theoretic problems. In this section, we shall provide a brief introduction to statistical equivalence, and show how it can be used to derive the minimax rates for the distributed multinomial model under bandwidth and differential privacy constraints. Further details on the statistical equivalence and a detailed proof are deferred Section A of the appendix.

Le Cam theory is a general framework for decision problems. At the core of this theory is the notion of a distance between statistical models, known as Le Cam's deficiency distance. The objective of this distance is to quantify the extent to which a complex statistical model can be approximated by a more simple one. If a model is close to another model in Le Cam's distance, then there is a mapping of solutions to decision theoretic problems from one model to the other. Whenever the risk of the decision problem is bounded, this means that similar performance can be achieved in the two models. Consequently, studying the complex model can be reduced to studying the corresponding simple model. For an extensive introduction to Le Cam theory, see e.g. [62, 81]. For a brief introduction; [63, 66].

Consider a model $\mathcal{P} = \{P_f : f \in \mathcal{F}\}$ (a collection of probability distributions) on a measurable space $(\mathcal{X}, \mathscr{X})$ (the sample space). For this article, we consider only models with Polish sample spaces and corresponding Borel sigma-algebras and dominated models, meaning that there exists a sigma-finite measure $\mu$ such that $P_f \ll \mu$ for all $f \in \mathcal{F}$. This greatly simplifies the definition of deficiency, given next.

Given another model $\mathcal{Q} = \{Q_f : f \in \mathcal{F}\}$ indexed by the same set $\mathcal{F}$ and sample space $(\tilde{\mathcal{X}}, \tilde{\mathscr{X}})$, we define the *deficiency of $\mathcal{P}$ with respect to $\mathcal{Q}$* as

$$\mathfrak{d}(\mathcal{P}; \mathcal{Q}) = \inf_C \sup_{f \in \mathcal{F}} \|P_f C - Q_f\|_{\mathrm{TV}}. \tag{12}$$

where the infimum is taken over all Markov kernels $C : \tilde{\mathscr{X}} \times \mathcal{X} \to [0, 1]$ and the probability measure $P_f C : \tilde{\mathscr{X}} \to [0, 1]$ is understood as $P_f C(A) := \int_{x \in \mathcal{X}} C(A|x) dP_f(x)$. This is equivalent to the more general notion of deficiency of [27] for dominated models on Polish spaces (see Proposition 9.2 in [68]).

Le Cam's deficiency distance between $\mathcal{P}$ and $\mathcal{Q}$ is then defined as $\Delta(\mathcal{P}, \mathcal{Q}) = \max\{\mathfrak{d}(\mathcal{P}; \mathcal{Q}), \mathfrak{d}(\mathcal{Q}, \mathcal{P})\}$. This semi-metric becomes a metric whenever $\mathcal{P}$ and $\mathcal{Q}$ are identified whenever $\mathfrak{d}(\mathcal{P}; \mathcal{Q}) + \mathfrak{d}(\mathcal{Q}, \mathcal{P}) = 0$. Two sequences of experiments $\mathcal{P}_\nu$ and $\mathcal{Q}_\nu$ are called *asymptotically equivalent* if their difference $\Delta(\mathcal{P}_\nu, \mathcal{Q}_\nu)$ tends to zero as $\nu$ approaches infinity. Conversely, such sequences shall be called *asymptotically nonequivalent* if $\Delta(\mathcal{P}_\nu, \mathcal{Q}_\nu) > c$ as $\nu \to \infty$ for a fixed constant $c > 0$.

In Section A.2, we prove that models that are close in the Le Cam metric (compared to $m$) have similar testing risks in the distributed setup. We leverage this result in combination with the fact that the distributed multinomial model is asymptotically equivalent to a $d$-dimensional distributed Gaussian model, which we describe next.

Consider for $q \in \mathcal{F}$ and $i = 1, \ldots, d$ the random variables

$$X_i^{(j)} = \sqrt{q_i} + \frac{1}{\sqrt{2n}} Z_i^{(j)} \quad \text{with} \quad Z^{(j)} = (Z_1^{(j)}, \ldots, Z_d^{(j)}) \sim N(0, I_d). \tag{13}$$

Let $P_f \equiv P_f^n$ denote the distribution of $X^{(j)} = (X_1^{(j)}, \ldots, X_d^{(j)})$. Let $\mathcal{P}$ denote the corresponding experiment. It is shown in [30] that $\mathcal{Q}$ is close to $\mathcal{P}$ in the Le Cam metric when $d$ is relatively small compared to $n$. More precisely, it follows from Theorem 1 and Section 7 in [30] that

$$\Delta(\mathcal{P}, \mathcal{Q}) \leq C_R \frac{d \log d}{\sqrt{n}}, \tag{14}$$

where $C_R > 0$ is a constant depending only on $R$. For the testing problem in Gaussian model, with hypotheses (4), the minimax rate can be derived using the results of [85] in case of bandwidth constraints and [22] in case of differential privacy constraints. The key tool from which the minimax rates can then be derived for the multinomial model is the following lemma, which allows comparison of the minimax testing risks for the multinomial and Gaussian models in regimes where the Le Cam distance is small. Its proof is given in Section A.2 of the appendix.

**Lemma 1.** *Suppose $m\Delta(\mathcal{Q}; \mathcal{P}) \leq \varrho$ for $\varrho > 0$. Then, it holds that*

$$\left| \inf_{T \in \mathscr{T}(\mathcal{P})} \mathcal{R}_{\mathcal{P}}(T, H_1) - \inf_{T \in \mathscr{T}(\mathcal{Q})} \mathcal{R}_{\mathcal{Q}}(T, H_1) \right| \leq 2\varrho,$$

*where $\mathscr{T}$ is either $\mathscr{T}_{SR}^b$, $\mathscr{T}_{LR}^b$, $\mathscr{T}_{SR}^{(\epsilon, \delta)}$ or $\mathscr{T}_{LR}^{(\epsilon, \delta)}$.*

# 5  Statistical non-equivalence of discrete and multivariate Gaussian distributions

Theorem 1 describing uniformity testing in the large sample regime and the result derived for $n = 1$, as displayed in (8), shows a striking difference terms of the role of the communication budget. Specifically, in the $n = 1$ regime, an exponential communication efficiency is observed, whereas in the large sample regime, the benefit is only linear. In this section, we shall provide some explanation for this phenomenon, and shall actually leverage this difference to show that the distributed multinomial model and the distributed Gaussian model are asymptotically non-equivalent: Two models are considered *asymptotically nonequivalent* if their Le Cam distance remains bounded away from zero, even as the amount of data increases in both models.

The multinomial model is equivalent to a model in which one observes $N^{(j)} = (N_1^{(j)}, \ldots, N_d^{(j)})$ taking values in $\{1, \ldots, n\}^d$, where $N_k^{(j)} \equiv N_k^{(j)}\left(\tilde{X}^{(j)}\right) = \left| \left\{ i : \tilde{X}_i^{(j)} = k \right\} \right|$. Let $\mathcal{Q}'$ denote the model generated by the observations $N^{(j)}$. This model is equivalent to $\mathcal{Q}$, meaning $\Delta(\mathcal{Q}, \mathcal{Q}') = 0$. To see this, note that for $x = (x_1, \ldots, x_n) \in \{1, \ldots, d\}^n$,

$$Q\left(\tilde{X}^{(j)} = x\right) = \prod_{i=1}^n Q\left(\tilde{X}_i^{(j)} = x_i\right) = \prod_{k \in \{1, \ldots, d\}} q_k^{|\{i : x_i = k\}|} \text{ for all } Q \in \mathcal{Q},$$

after which the aforementioned equivalence follows by the Neyman-Fisher factorization criterion, e.g. Lemma 2 in the appendix. When $n$ is large compared to $d$, one could standardize the count statistics $N^{(j)}$ to obtain a statistic that tends towards a $d$-dimensional Gaussian random vector. When $d$ and $m$ are not too large with respect to $n$, one can obtain transcripts and corresponding test statistics from these approximately Gaussian vectors, that resemble those one would consider in the Gaussian model, and attain the corresponding minimax rates.

Since the observation $N^{(j)}$ takes values in $\{1, \ldots, n\}^d$, the full data can be transmitted whenever there are at least $d \log_2 n$-bits are available per server. However, recalling that the observation $\tilde{X}^{(j)}$ takes values in the space $\{1, \ldots, d\}^n$, which has cardinality bounded above by $2^{n \log_2 d}$, we also obtain that the full data can be transmitted whenever $n \log_2 d$-bits are available. Consequently, whenever

$$b \gtrsim d \log_2(n+1) \wedge n \log_2 d,$$

the distributed problem has the same minimax separation rate for the hypothesis in (4) as the unconstrained problem with $nm$ observations; $\rho_{\mathcal{Q}}^2 \asymp \frac{\sqrt{d}}{mn}$. For the Gaussian problem, this is only the case whenever $b \gtrsim d$, as can be seen from Theorem 4. This indicates a kind of "tipping point" occuring whenever $n$ gets small compared to $d$, where in a bandwidth constraint distributed setting, the testing problem in for the Gaussian model starts to exhibit very different behavior.

Interestingly, this does not imply that the multinomial model is "easier" from a distributed testing under bandwidth constraints perspective, as there are sub-regimes in which the Gaussian model has a solution whereas the multinomial model does not and vice versa. It indicates that the "communication complexity" of the sample space matters in the respective decision problems. We can leverage this fact to obtain a lower-bound on the Le Cam distance between the multinomial model and the Gaussian model; which is the content of the next theorem.

**Theorem 3.** *There exists constants $C > 0$ and $c > 0$ such that for any $n, d \in \mathbb{N}$ with*

$$\frac{d}{n \log(d)} \geq C \quad and \quad n \geq \sqrt{d} \log(d), \tag{15}$$

*it holds that*

$$\mathfrak{d}(\mathcal{Q}, \mathcal{P}) \geq c, \tag{16}$$

*where $\mathcal{P}$ is the experiment generated by the observations in* (13)*, $\mathcal{Q}$ is generated according to* (3)*, both indexed by $\mathcal{F}$ as given in* (5)*.*

The proof of the theorem is given in Section D. It leverages that there exist distributed, $b$-bit bandwidth constraint settings in which the (distributed) multinomial model allows for consistent goodness-of-fit testing, whereas the (distributed) Gaussian model does not. The result then readily follows from the distributed equivalence results derived in Section A.2. The fact that the separation in the respective (distributed) testing risks occurs for a constant number of servers, yields that the two models are asymptotically nonequivalent whenever $\sqrt{d}/\log^2(d) \geq d/n \gg 1$. This reasoning crucially exploits the differing minimax rates that occur under the bandwidth constraint, since without such a constraint, the same goodness-of-fit testing problem of (4) would have similar minimax performance for both of the models.

## 6   Discussion

We have derived minimax separation rates for uniformity testing in the distributed multinomial model under bandwidth and differential privacy constraints, in the large sample regime where $md \log d/\sqrt{n} = o(1)$. When contrasted with existing results for large sample regimes, the minimax rates show that the large sample regime is subject to distinctly different phenomena.

The applicability of our results is somewhat constrained by the requirement that $md \log d/\sqrt{n} = o(1)$, which limits the range of model characteristics we can consider. Consequently, further work is needed to understand the behavior of the distributed multinomial model in other regimes. The non-equivalence result in Theorem 3 indicates that the distributed multinomial model and the distributed Gaussian model are fundamentally different regarding distributed statistical decision problems when the sample size is small. Therefore, direct analysis of the distributed multinomial model might be necessary, requiring new techniques to derive minimax rates. We note, however, that this pertains to the specific Gaussian model formulated in (13), and there might be a different Gaussian model that is equivalent to the distributed multinomial model even in the small $n$ regime.

The results in this paper are derived through the notion of statistical equivalence, which is a powerful tool for establishing minimax rates in statistical decision theoretic problems. The results and techniques can be applied more generally to other distributed inference problems, and proving more general results concerning statistical equivalence and distributed inference is an interesting avenue for future research.

A downside of leveraging statistical equivalence is that it generally does not provide a direct path to obtain methods that are minimax rate optimal. However, Theorem 1 and Section 7 in [30] provide a specific transformation that converts the local multinomial sample into a statistic approximately distributed as a Gaussian random vector. Such a transformation, combined with the rate optimal methods given in [85] and [22], provide guidance to construct methods that attain the minimax rates described in this article.

## Acknowledgments and Disclosure of Funding

The author is grateful to the anonymous reviewers for their valuable feedback and suggestions and to Aad van der Vaart for his thorough reading of an earlier draft.

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

# A  Le Cam theory in distributed setting

We introduce some formal notions of Le Cam theory first in Section A.1. Then, in Section A.2, we study the equivalence of models in the distributed setting. The theoretical developments presented in this section apply to general models denoted by $\mathcal{P}$ and $\mathcal{Q}$; although the main text specifically focuses on the Gaussian location model for $\mathcal{P}$ and the multinomial model for $\mathcal{Q}$, the machinery developed here is applicable to general statistical models.

## A.1  Preliminary notions of Le Cam theory

A *statistical experiment* is a set of probability distributions $\mathcal{P} = \{P_f : f \in \mathcal{F}\}$ (a model) on a measurable space $(\mathcal{X}, \mathscr{X})$ (the sample space). For the purpose of simplification, we shall consider only statistical experiments with Polish sample spaces and corresponding Borel sigma-algebras. Furthermore, we shall only consider dominated models, meaning that there exists a sigma-finite measure $\mu$ such that $P_f \ll \mu$ for all $f \in \mathcal{F}$. In a slight abuse of terminology, we shall sometimes refer to $\mathcal{P}$ as the experiment, suppressing the presence of the sample space and indexing set.

Given another statistical experiment with model $\mathcal{Q} = \{Q_f : f \in \mathcal{F}\}$ indexed by the same set $\mathcal{F}$ and sample space $(\tilde{\mathcal{X}}, \tilde{\mathscr{X}})$, we define the *deficiency of $\mathcal{P}$ with respect to $\mathcal{Q}$* as

$$\mathfrak{d}(\mathcal{P}; \mathcal{Q}) = \inf_C \sup_{f \in \mathcal{F}} \|P_f C - Q_f\|_{\mathrm{TV}}. \tag{17}$$

where the infimum is taken over all Markov kernels $C : \tilde{\mathscr{X}} \times \mathcal{X} \to [0,1]$ and the probability measure $P_f C : \tilde{\mathscr{X}} \to [0,1]$ is understood as

$$P_f C(A) := \int_{x \in \mathcal{X}} C(A|x) dP_f(x). \tag{18}$$

This is equivalent to the more general notion of deficiency of [27] for dominated models on Polish spaces (see Proposition 9.2 in [68]).

The deficiency $\mathfrak{d}(\mathcal{P}; \mathcal{Q})$ quantifies the degree to which $\mathcal{Q}$ can be approximated by an experiment $\mathcal{P}$. If $\mathfrak{d}(\mathcal{P}; \mathcal{Q}) \leq \varrho$, it implies that for bounded loss functions, each decision procedure within $\mathcal{Q}$ has an associated procedure in $\mathcal{P}$ that achieves nearly the same risk, up to a multiple of $\varrho$.

To make this precise, let $\mathcal{F}$ be a measurable space and consider a function $\ell : \mathcal{F} \times \mathcal{D} \to [0,1]$ on a measurable space $(\mathcal{D}, \mathscr{D})$, such that $t \mapsto \ell(f, t)$ is measurable for all $f \in \mathcal{F}$, which we shall refer to a *loss functions*. We shall consider a *decision procedure* for $(\mathcal{Q}, \mathcal{D})$ to be a Markov kernel $D : \mathscr{D} \times \tilde{\mathcal{X}} \to [0,1]$. If $\mathfrak{d}(\mathcal{P}; \mathcal{Q}) \leq \varrho$, there exists $C : \tilde{\mathscr{X}} \times \mathcal{X} \to [0,1]$ such that for all decision procedures $D$ for $(\mathcal{Q}, \mathcal{D})$ we have that

$$\int \ell(f, \varphi) dP_f CD(\varphi) \leq \int \ell(f, \varphi) dQ_f D(\varphi) + 2\varrho, \quad \text{for all } f \in \mathcal{F}.$$

Here, the Markov kernel $Q_f D$ is to be understood in the sense of (18) and $CD : \mathscr{D} \times \mathcal{X} \to [0,1]$ as

$$CD(A|x) = \int D(A|\tilde{x}) dC(\tilde{x}|x).$$

There is also the following reverse implication; suppose that there exists a loss function $\ell : \mathcal{F} \times \mathcal{D} \to [0,1]$ on a measurable space $(\mathcal{D}, \mathscr{D})$, and

$$\inf_C \inf_D \sup_{f \in \mathcal{F}} \left| \int \ell(f, \varphi) dQ_f D(\varphi) - \int \ell(f, \varphi) dP_f CD(\varphi) \right| > 2\varrho,$$

where the two infimums are over all decision procedures $D$ and Markov kernels $C : \tilde{\mathscr{X}} \times \mathcal{X} \to [0,1]$. Then, $\mathfrak{d}(\mathcal{Q}, \mathcal{P}) > \varrho$. This follows immediately from e.g. Lemma 12 in the appendix, since $x \mapsto \int \ell(f, \varphi) dD(\varphi|x)$ is measurable. In the more extensive framework considered in e.g. [27], such a reverse implication for risk functions fully characterizes the deficiency between two models, but this framework is not needed in what follows.

Le Cam's deficiency distance between $\mathcal{P}$ and $\mathcal{Q}$ is then defined as

$$\Delta(\mathcal{P}, \mathcal{Q}) = \max \{\mathfrak{d}(\mathcal{P}; \mathcal{Q}), \mathfrak{d}(\mathcal{Q}, \mathcal{P})\}.$$

This semi-metric becomes a metric whenever $\mathcal{P}$ and $\mathcal{Q}$ are identified whenever $\mathfrak{d}(\mathcal{P};\mathcal{Q}) + \mathfrak{d}(\mathcal{Q},\mathcal{P}) = 0$. Two sequences of experiments $\mathcal{P}_\nu$ and $\mathcal{Q}_\nu$ are called *asymptotically equivalent* if their difference $\Delta(\mathcal{P}_\nu, \mathcal{Q}_\nu)$ tends to zero as $\nu$ approaches infinity. Conversely, such sequences shall be called *asymptotically nonequivalent* if $\Delta(\mathcal{P}_\nu, \mathcal{Q}_\nu) > c$ as $\nu \to \infty$ for a fixed constant $c > 0$.

The final notion we shall recall is that of sufficiency. A statistic $S : \mathcal{X} \to \tilde{\mathcal{X}}$ is *sufficient for the model* $\mathcal{P}$ if for any $A \in \mathscr{X}$ there exists a measurable map $\psi_A : \tilde{\mathcal{X}} \to \mathbb{R}$ such that

$$P_f\left(A \cap S^{-1}(B)\right) = \int_B \psi_A(\tilde{x}) dP_f^S(\tilde{x}) \ \text{ for all } B \in \tilde{\mathscr{X}} \ \text{ and } f \in \mathcal{F}.$$

Here, the measure $P_f^S$ is to be understood as the push-forward measure $P_f^S(B) = P_f(S^{-1}(B))$. A sufficient statistic allows for transforming observations from one model to another, "sufficient" model which is equivalent in the sense of Le Cam distance. That is, if $S$ is a sufficient statistic for $\mathcal{P}$, then the model $\mathcal{P}' := \{P_f^S : f \in \mathcal{F}\}$ satisfies $\Delta(\mathcal{P}, \mathcal{P}') = 0$.

The next lemma is the Neyman-Fisher factorization theorem gives a useful characterization of sufficiency of a statistic for models that admit densities with respect to the same dominating measure.

**Lemma 2.** *Suppose that $P_f \ll \mu$ for all $P_f \in \mathcal{P}$ with $\mu$ a sigma-finite measure. A statistic $S : \mathcal{X} \to \tilde{X}$ is sufficient for $\mathcal{P}$ if and only if there exists measurable functions $g_f : \mathbb{R} \to \mathbb{R}$ and $h : \mathcal{X} \to \mathbb{R}$ such that*

$$\frac{dP_f}{d\mu}(x) = g_f(S(x))h(x) \ \text{ for almost every } x \in \mathcal{X} \text{ and every } f \in \mathcal{F}. \tag{19}$$

A proof for both the lemma and the last statement of the previous paragraph can be found in Chapter 5 of [62].

## A.2 Equivalence of distributed decision problems

We now turn to the distributed setting considered in the paper, where $j = 1, \ldots, m$ servers each receive data $X^{(j)}$ drawn from a distribution $P_f$ and sample space $(\mathcal{X}, \mathscr{X})$. Each of the servers communicates a transcript based on the data to a central server, which based on the aggregated transcripts computes its solution to the decision problem at hand.

The tools developed in this section apply to wider range of distributed architectures than the one considered in the main text of the paper, as introduced in Section 2. The framework introduced here accommodates various forms of interaction between servers, including sequential and blackboard protocols (see e.g. [6, 16]). In contrast, the main text focuses on servers that either do not communicate (local randomness protocols) or utilize a shared randomness source (a special case of sequential or blackboard communication).

A *distributed protocol for the experiment* $\mathcal{P}$ with decision space $(\mathcal{D}, \mathscr{D})$ consists of a triplet $\{D, \mathcal{K}, (\mathcal{U}, \mathscr{U}, \mathbb{P}^U)\}$, a Markov kernel $D : \mathscr{D} \times \bigotimes_{j=1}^m \mathcal{Y}^{(j)} \to [0,1]$ and a probability space $(\mathcal{U}, \mathscr{U}, \mathbb{P}^U)$, and $\mathcal{K}$ is a collection of Markov kernels.

To unpack all this notation: the Markov kernel $D$ takes the role of the decision procedure, where the decision is to made on the basis of the transcripts generated by the Markov kernels $\mathcal{K}$. The transcripts are in turn generated based on the data and a source of shared randomness independent of the data. The probability space $(\mathcal{U}, \mathscr{U}, \mathbb{P}^U)$ plays the role of the source of randomness that is shared by the servers. The distributed protocol is said to have *no access to shared randomness* or to be *a local randomness protocol* if $\mathscr{U}$ is the trivial sigma-algebra.

In this section, we shall consider three types of communication architectures:

- *One shot protocols*: $\mathcal{K} = \{K^j\}_{j=1,\ldots,m}$ where $K^j : \mathcal{Y}^{(j)} \times (\mathcal{X} \times \mathcal{U}) \to [0,1]$. These protocols are what are considered in the main text of the paper.
- *Sequential protocols*: $\mathcal{K} = \{K^j\}_{j=1,\ldots,m}$ where $K^j : \mathcal{Y}^{(j)} \times (\mathcal{X} \times \mathcal{U} \times \mathcal{Y}^{(1)} \times \cdots \times \mathcal{Y}^{(j-1)}) \to [0,1]$. That is, the transcript generated by server $j$ is based on the data, the shared randomness and the transcripts of the previous servers.
- *Blackboard protocols*:

- *Blackboard protocols*: $\mathcal{K} = \{K_t^j\}_{j=1,\ldots,m,t=1,\ldots,T}$ where $K_1^j : \mathscr{Y}^{(j)} \times (\mathcal{X} \times \mathcal{U}) \to [0,1]$ and

$$K_t^j : \mathscr{Y}^{(j)} \times \left( \mathcal{X} \times \mathcal{U} \times (\mathcal{Y}^{(1)} \times \cdots \times \mathcal{Y}^{(m)})^{\otimes(t-1)} \right) \to [0,1],$$

for $t = 2, \ldots, T$. That is, the transcript generated by server $j$ is based on the data, the shared randomness, and the transcripts of *all* the servers from the previous round.

For one shot protocols, we have in terms of random variables that $X^{(j)} \sim P_f$, $U \sim \mathbb{P}^U$, $Y^{(j)}|(X^{(j)}, U) \sim K^j(\cdot|X^{(j)}, U)$ for $j = 1, \ldots, m$ and $\varphi \sim D(\cdot|Y)$ with $Y = (Y^{(1)}, \ldots, Y^{(m)})$. This gives rise to a Markov chain

$$
\begin{array}{ccccc}
X^{(1)} & \longrightarrow & Y^{(1)}|U & \searrow & \\
\vdots & \longrightarrow & \vdots & \longrightarrow & \varphi. \\
X^{(m)} & \longrightarrow & Y^{(m)}|U & \nearrow &
\end{array}
\tag{20}
$$

For $x = (x^{(1)}, \ldots, x^{(m)}) \in \mathcal{X}^m$, $u \in \mathcal{U}$ and $\{K^j\}_{j=1,\ldots,m}$, let $x \mapsto K(A|x)$ be the Markov kernel product distribution $\bigotimes_{j=1}^m K^j(\cdot|x^{(j)}, u)$. Given a distributed protocol and i.i.d. data from $P_f$ we shall use $\mathbb{P}_f$ to denote the joint distribution the data $X \sim P_f^m$, the shared randomness $U \sim \mathbb{P}^U$ and $Y = (Y^{(1)}, \ldots, Y^{(m)})$ with $Y|(X, U) \sim K(Y|X, U)$. We have that $P_f^m K = \bigotimes_{j=1}^m P_f K^j$ and the push-forward measure of $Y$ then disintegrates as

$$\mathbb{P}_f^Y(A) = P_f^m \mathbb{P}^U K(A) = \mathbb{P}^U P_f^m K(A) = \int d\bigotimes_{j=1}^m P_f K^j(\cdot|X^{(j)}, u)(A) d\mathbb{P}^U(u), \tag{21}$$

where the second equality follows from the independence of $U$ with the data $X := (X^{(1)}, \ldots, X^{(m)})$ drawn from $P_f$.

For sequential protocols, the push-forward measure of $Y$ instead disintegrates as

$$\mathbb{P}_f^Y(A) = \int_{\mathcal{U}} \left[ \int \cdots \int \mathbb{1}_A(y) dP_f K^m \left( y^m \mid X^{(m)}, u, (y)_{j=1}^{m-1} \right) \cdots dP_f K^1(y^1 \mid X^{(1)}, u) \right] d\mathbb{P}^U(u), \tag{22}$$

For blackboard protocols, a similar disintegration applies for each of the rounds.

A one shot or sequential distributed protocol is said to satisfy a *b-bit bandwidth constraint* if its kernels $\{K^j\}_{j=1,\ldots,m}$ are defined on spaces satisfying $|\mathcal{Y}^{(j)}| \leq 2^b$. For blackboard protocols, various bandwidth constraints can be imposed, such as a $b$-bit bandwidth constraint for each round $t = 1, \ldots, T$.

Given Markov Kernels $C^j : \mathscr{X} \times \tilde{\mathcal{X}} \to [0,1]$, $j = 1, \ldots, m$, a distributed one shot protocol $\{D, \{K^j\}_{j=1,\ldots,m}, (\mathcal{U}, \mathscr{U}, \mathbb{P}^U)\}$ for the model $\mathcal{P}$, yields a distributed protocol for the model $\mathcal{Q}$: $\{D, \{CK^j\}_{j=1,\ldots,m}, (\mathcal{U}, \mathscr{U}, \mathbb{P}^U)\}$. If $\{K^j\}_{j=1,\ldots,m}$ is $b$-bit bandwidth constraint, the collection of kernels $\{C^j K^j\}_{j=1,\ldots,m}$ do so too, as each $C^j K^j$ is defined on $\mathscr{Y}^{(j)} \times \tilde{\mathcal{X}}$.

Similarly, for a sequential protocol, the Markov kernels $C^j$ and $K^j$ yield a distributed sequential protocol for the model $\mathcal{Q}$ with kernels $\{C^j K^j\}_{j=1,\ldots,m}$. If each $K^j$ is $b$-bit bandwidth constraint, so is each $C^j K^j$. For blackboard protocols, the same reasoning applies to each round $t = 1, \ldots, T$. That is, type of protocol defined by the kernels $\mathcal{K}$ is "closed under composition" with kernels $C^j$ between $\tilde{\mathcal{X}}$ with target space $\mathcal{X}$, where bandwidth constraints are preserved.

We shall consider the notion of local $\epsilon$-differential privacy of Definition 3. A Markov kernel $K : \mathscr{Y} \times \mathcal{X} \to [0,1]$ is called *locally $(\epsilon, \delta)$-differentially private* if

$$K(A|x) \leq e^\epsilon K(A|x') + \delta \quad \text{for all } A \in \mathscr{Y} \text{ and } x, x' \in \mathcal{X}. \tag{23}$$

Since the definition of differential privacy depends heavily on what one defines as the sample space, it is difficult to obtain a similar "transfer of distributed protocols" that respects the $(\epsilon, \delta)$-differential privacy constraint, hence the choice to consider local constraints only.

A one shot or sequential distributed protocol shall be called locally $\epsilon$-differentially private if (23) holds for each $K^j$; $j = 1, \ldots, m$. For blackboard protocols, one can impose a $(\epsilon, \delta)$-differential privacy constraint for each round $t = 1, \ldots, T$, or for the entire output over $t = 1, \ldots, T$ rounds. The following lemma shows that local $\epsilon$-differential privacy, just like bandwidth constraints, carry over from one model to the other.

**Lemma 3.** *Let $(\mathcal{X}, \mathscr{X})$ and $(\tilde{\mathcal{X}}, \tilde{\mathscr{X}})$ be measurable spaces and consider Markov kernels $C :$ $\tilde{\mathscr{X}} \times \mathcal{X} \to [0, 1]$ and $K : \mathscr{Y} \times \mathcal{X} \to [0, 1]$. If $K$ is $b$-bit bandwidth constraint, so is the Markov kernel $CK : \mathscr{Y} \times \tilde{\mathcal{X}} \to [0, 1]$. If $K$ is locally $\epsilon$-differentially private, so is $CK$. Furthermore, for any collection of Markov kernels $\mathcal{K}$, the same reasoning applies to the collection $\{CK\}_{K \in \mathcal{K}}$, preserving bandwidth constraints and local differential privacy, as well as the protocol's architecture.*

*Proof.* The first statement has been remarked on earlier in the section. For the second statement, consider arbitrary $\tilde{x}, \tilde{x}' \in \tilde{\mathcal{X}}$ and $A \in \mathscr{Y}$. Using that $C$ is a Markov kernel and applying (23) to $K$ yields

$$CK(A|\tilde{x}) = \int K(A|x) dC(x|\tilde{x}) = \int \int K(A|x) dC(x|\tilde{x}) dC(x'|\tilde{x}')$$

$$\leq e^{\epsilon} \int K(A|x') dC(x'|\tilde{x}') + \delta = e^{\epsilon} CK(A|\tilde{x}') + \delta,$$

which shows $CK$ is locally $(\epsilon, \delta)$-differentially private. The above argument applies pointwise for all other conditional arguments in the Markov kernel, hence the same reasoning applies to shared randomness, sequential and blackboard protocols. $\square$

In an abuse of notation, let $D$ denote the entire distributed protocol (triplet)

$$\{D, \{K^j\}_{j=1,\ldots,m}, (\mathcal{U}, \mathscr{U}, \mathbb{P}^U)\}$$

for the experiment $\mathcal{P}$ (indexed by $\mathcal{F}$) with decision space $(\mathcal{D}, \mathscr{D})$. Given $D$ and a loss function $\ell : \mathcal{F} \times \mathcal{D} \to [-1, 1]$, we define the *distributed risk of $D$ in $\mathcal{P}$ for $\ell$* as

$$\mathcal{R}_{\mathcal{P}}(D, \ell) := \sup_{f \in \mathcal{F}} \int \int \int \ell(f, \varphi) dD(\varphi|y) \, d\bigotimes_{j=1}^{m} P_f K^j(\cdot|X^{(j)}, u)(y) d\mathbb{P}^U(u).$$

We are now ready to formulate a straightforward consequence for the distributed risk, following from models being close in Le Cam distance. This finding, formulated in Lemma 4, shall serve as one of the main tools for deriving the main results. It states roughly that, whenever there is a $b$-bit bandwidth constrained distributed protocol that achieves a certain risk is one model and there is small deficiency with the other model relative to the number of servers, there exists a $b$-bit distributed protocol that achieves comparable risk for the other model. A similar statement holds under local differential privacy constraints. If there is a locally $(\epsilon, \delta)$-differentially private distributed procedure in the one model and there is small deficiency with another model, it means that there is comparable risk for the privacy constraint distributed decision problem.

**Lemma 4.** *Let $m \in \mathbb{N}$. Consider two experiments $\mathcal{P}$ and $\mathcal{Q}$ with indexing set $\mathcal{F}$, satisfying $m\mathfrak{d}(\mathcal{Q}; \mathcal{P}) \leq \varrho$ for some $\varrho > 0$. Let $\mathscr{J}_{\mathcal{P}}$ and $\mathscr{J}_{\mathcal{Q}}$ denote the class of $b$-bit bandwidth constraint shared randomness protocols for the models $\mathcal{P}$ and $\mathcal{Q}$ respectively.*

*Then, for any loss function $\ell : \mathcal{F} \times \mathcal{D} \to [0, 1]$,*

$$\inf_{D \in \mathscr{J}_{\mathcal{Q}}} \mathcal{R}_{\mathcal{Q}}(D, \ell) - \inf_{D \in \mathscr{J}_{\mathcal{P}}} \mathcal{R}_{\mathcal{P}}(D, \ell) \leq \varrho.$$

*where in the infimum, in an abuse of notation, $D$ denotes the entire distributed protocol triplet $\{D, \{K^j\}_{j=1,\ldots,m}, (\mathcal{U}, \mathscr{U}, \mathbb{P}^U)\}$.*

*The same statement is holds for $\mathscr{J}_{\mathcal{P}}$ and $\mathscr{J}_{\mathcal{Q}}$ denoting either classes of $b$-bit bandwidth constraint local randomness, sequential protocols, or any of these distributed protocols satisfying local $(\epsilon, \delta)$-differential privacy constraints, for the respective models $\mathcal{P}$ and $\mathcal{Q}$. If $Tm\mathfrak{d}(\mathcal{Q}; \mathcal{P}) \leq \varrho$, the same statement holds for blackboard protocols with $T$ rounds.*

*Remark* 1. This exemplifies also that, even though models $\mathcal{P}^m = \{P_f^m : f \in \mathcal{F}\}$ and $\mathcal{Q}^m = \{Q_f^m : f \in \mathcal{F}\}$ are close in Le Cam distance, distributed decision problems formulated in terms the models $\mathcal{P}$ and $\mathcal{Q}$, can have greatly different performance in terms of associated risks.

*Proof.* By e.g. Theorem 2 in [62], $m\mathfrak{d}(\mathcal{Q};\mathcal{P}) \leq \varrho$ implies that there exists a kernel $C : \mathscr{X} \times \tilde{\mathscr{X}} \to [0,1]$ such that

$$\sup_{f \in \mathcal{F}} \|P_f - Q_f C\|_{\mathrm{TV}} \leq \varrho/m. \tag{24}$$

By Lemma 3, the kernels $\tilde{\mathcal{K}} = \{CK : K \in \mathcal{K}\}$ satisfy the a $b$-bit bandwidth constraint or local $(\epsilon, \delta)$-differential privacy constraint if the collection $\{K^j\}_{j=1,\ldots,m}$ does. To illustrate this further, consider a one shot distributed protocol for $\mathcal{P}$, $\{D, \{K^j\}_{j=1,\ldots,m}, (\mathcal{U}, \mathscr{U}, \mathbb{P}^U)\} \in \mathscr{J}_{\mathcal{P}}$, the distributed protocol $\tilde{D} = \{D, \{CK^j\}_{j=1,\ldots,m}, (\mathcal{U}, \mathscr{U}, \mathbb{P}^U)\}$ is then an element of $\mathscr{J}_{\mathcal{Q}}$.

Using the fact that $\ell$ is bounded by one and Lemma 13 in the appendix, it follows that

$$\mathcal{R}_{\mathcal{Q}}(\tilde{D}, \ell) - \mathcal{R}_{\mathcal{P}}(D, \ell) \leq \|\mathbb{P}^U \bigotimes_{j=1}^m P_f K^j - \mathbb{P}^U \bigotimes_{j=1}^m Q_f CK^j\|_{\mathrm{TV}}$$

$$\leq \sum_{j=1}^m \|\mathbb{P}^U P_f K^j - \mathbb{P}^U Q_f CK^j\|_{\mathrm{TV}}.$$

By Lemma 14 in the appendix,

$$\|\mathbb{P}^U P_f K^j - \mathbb{P}^U Q_f CK^j\|_{\mathrm{TV}} \leq \|\mathbb{P}^U P_f - \mathbb{P}^U Q_f C\|_{\mathrm{TV}} = \|P_f - Q_f C\|_{\mathrm{TV}},$$

which combined with (24) finishes for one shot protocols.

Similar reasoning applies to sequential protocols. We start by noting that

$$\|P_f K^1(\cdot \mid u) - Q_f CK^1(\cdot \mid u)\|_{\mathrm{TV}} \leq \|P_f - Q_f C\|_{\mathrm{TV}} \leq \varrho/m,$$

Lemma 11 implies that (for any $u$) there exists a coupling $\mathbb{P}$ of $Y^{(1)} \sim P_f K^1(\cdot \mid u)$ and $\tilde{Y}^{(1)} \sim Q_f CK^1(\cdot \mid u)$ such that $2\mathbb{P}(Y^{(1)} \neq \tilde{Y}^{(1)}) \leq \varrho/m$. Write

$$K^{m:2}(\cdot \mid x^m, \ldots, x^2, u, y^1) := \int \cdots \int dK^m(y^m \mid x^m, u, y^{m-1}, \ldots, y^1) \cdots dK^2(y^2 \mid x^2, u, y^1).$$

Using the disintegration relationship of (22), we obtain

$$\|\mathbb{P}^U P_f^m K - \mathbb{P}^U Q_f CK\|_{\mathrm{TV}} \leq \int \|P_f^m K(\cdot \mid u) - Q_f CK(\cdot \mid u)\|_{\mathrm{TV}} d\mathbb{P}^U(u)$$

$$\leq \varrho/m + \int \mathbb{E}_f^{Y^{(1)}|U=u} \|P_f^m K^{m:2}(\cdot \mid u, Y^{(1)}) - Q_f CK^{m:2}(\cdot \mid u, Y^{(1)})\|_{\mathrm{TV}} d\mathbb{P}^U(u).$$

Iterating the above argument $m - 1$ times, we the statement for sequential protocols. The statement for blackboard protocols follows by combining the above arguments for each round $t = 1, \ldots, T$. $\square$

In the remainder of this text, we shall constrain ourselves to a particular bounded risk function and distributed decision problem; distributed hypothesis testing. The following corollary formalizes the statement at the start of the paragraph for the testing a simple null versus a compositive alternative hypothesis in the distributed setting. To that extent, consider a test of the hypotheses

$$H_0 : f = f_0 \quad \text{versus the alternative hypothesis} \quad f \in H_1 \tag{25}$$

and an experiment $\mathcal{P}$ with indexing set $\mathcal{F}$ satisfying $\{f_0\} \cup H_1 \subset \mathcal{F}$. Consider for $m \in \mathbb{N}$ a *distributed testing protocol* for the model $\mathcal{P}$ to be a distributed protocol $T \equiv \{D, \{K^j\}_{j=1,\ldots,m}, (\mathcal{U}, \mathscr{U}, \mathbb{P}^U)\}$, where in a slight abuse of notation, we shall also use $T$ to denote the (possibly randomized) test $T|Y \sim D(\cdot|Y)$. Recalling the notation $\mathbb{P}_f^Y = P_f^m \mathbb{P}^U K$ as given in (21), define the distributed testing risk for the hypotheses in (25) and the model $\mathcal{P}$ as

$$\mathcal{R}_{\mathcal{P}}(T, H_1) := \mathbb{P}_{f_0}^Y D(T|Y) + \sup_{f \in H_1} \mathbb{P}_f^Y (1 - D(T|Y)).$$

Here, $D(T|Y) := D(\{1\}|Y)$, but one can equivalently consider a deterministic measurable map $T : \prod_{j=1}^{m} \mathcal{Y}^{(j)} \to [0, 1]$ without loss of generality. Let $\mathscr{T}_{\text{SR}}^{b}(\mathcal{P})$ (resp. $\mathscr{T}_{\text{LR}}^{b}(\mathcal{P})$) denote the set of shared randomness (resp. local randomness) distributed testing protocols for $\mathcal{P}$ satisfying a $b$-bit bandwidth constraint. Similarly, let $\mathscr{T}_{\text{SR}}^{(\epsilon,\delta)}(\mathcal{P})$ (resp. $\mathscr{T}_{\text{LR}}^{(\epsilon,\delta)}(\mathcal{P})$) denote the set of shared randomness (resp. local randomness) distributed testing protocols for $\mathcal{P}$ satisfying a local $(\epsilon, \delta)$-differential privacy constraint. Define the same classes for the model $\mathcal{Q}$ in the obvious way. Using Lemma 4, we obtain the following result, which is a more general version of Lemma 1.

**Lemma 5.** *Consider experiments $\mathcal{P}, \mathcal{Q}$ such that $m\mathfrak{d}(\mathcal{Q}; \mathcal{P}) \leq \varrho$ for $\varrho > 0$. It holds that*

$$\inf_{T \in \mathscr{T}(\mathcal{P})} \mathcal{R}_{\mathcal{P}}(T, H_1) \leq \inf_{T \in \mathscr{T}(\mathcal{Q})} \mathcal{R}_{\mathcal{Q}}(T, H_1) + 2\varrho,$$

*where $\mathscr{T}$ is either $\mathscr{T}_{SR}^{b}, \mathscr{T}_{LR}^{b}, \mathscr{T}_{SR}^{(\epsilon,\delta)}$ or $\mathscr{T}_{LR}^{(\epsilon,\delta)}$.*

*Proof.* Given $\{T, \{K^j\}_{j=1,\dots,m}, (\mathcal{U}, \mathscr{U}, \mathbb{P}^U)\} \in \mathscr{T}(\mathcal{P})$, Lemma 4 applied to the loss function

$$\ell(f, t) := t\mathbb{1}_{\{f_0\}}(f) + (1 - t)\mathbb{1}_{H_1}(f)$$

and using that $\{f_0\} \cup H_1 \subset \mathcal{F}$ gives

$$\mathbb{P}_{f_0}^{Y} D(T|Y) < \mathbb{Q}_{f_0}^{Y} D(T|Y) + \varrho \quad \text{and} \quad \sup_{f \in H_1} \mathbb{P}_f^{Y}\left(1 - D(T|Y)\right) < \sup_{f \in H_1} \mathbb{Q}_f^{Y}\left(1 - D(T|Y)\right) + \varrho$$

for some distributed testing protocol $\{D, \{\tilde{K}^j\}_{j=1,\dots,m}, (\mathcal{U}, \mathscr{U}, \mathbb{P}^U)\}$ in $\mathscr{T}(\mathcal{Q})$, which yields the first statement. The second statement follows by symmetry of the argument. $\square$

The implications of Lemma 4 have implications beyond the testing framework. Whilst in distributed estimation settings, the loss function under consideration is typically not bounded, rates can still be derived in probability. That is, if the minimax rate for the distance $d$ on $\mathcal{F}$ in the model $\mathcal{P}_\nu$ is $\rho_\nu$, the bounded loss function

$$\ell_\nu(f, g) = \mathbb{1}\left\{d(f, g) \leq C\rho_\nu\right\} \quad \text{for } C > 0$$

can be used describe minimax estimation rates (in probability) between models $\mathcal{P}$ and $\mathcal{Q}$. Since the paper is about testing, we shall not pursue this direction any further beyond this remark.

In the next sections, we will explore the consequences of Lemma 5 for minimax distributed testing rates for both bandwidth- and privacy constraints.

# B   Difficulties in direct analysis of the multinomial model under information constraints

Lower-bounds for both estimation and testing problems are typically established by bounding divergence measures between probability distributions, such as the chi-square divergence, mutual information, or total variation; [3, 6, 7, 16**?** , 49, 9, 99, 13, 84, 85, 23, 22].

The proof techniques used for discrete distribution estimation employed in [3, 6] and [7], tight lower-bounds can often be obtained by "tensorizing" the divergence—breaking the problem into a sum of local divergences. The inferential cost incurred due to bandwidth or privacy constraints are then captured via data processing arguments. Similar tensorization techniques are employed in other estimation problems, see for example [16, 99].

However, this tensorization approach does not yield tight bounds for testing problems. For example, [84] uses mutual information in a tensorization framework for testing but only recovers optimal rates when each server communicates a single bit. Similarly, [8] attempts an estimation-based approach for goodness-of-fit testing but obtains tight lower-bounds only under 1-bit constraints, as detailed in Section 4 of their paper.

To achieve tight lower-bounds in testing problems, especially under communication and privacy constraints, different techniques are required. Papers [9, 85, 22] employ methods that significantly diverge from those used in estimation. In [9], a combinatorial expansion of the likelihood is used, effective for small sample sizes in the multinomial model but not generalizable to large numbers of

observations. [85] and [22] address this limitation in the Gaussian setting by utilizing the Brascamp-Lieb inequality from functional analysis, which explicitly leverages the Gaussian properties of the log-likelihood. This approach is not directly applicable to discrete models, due to the lack of quadratic structure in the log-likelihood of the multinomial model.

## C  Separation rates for the Gaussian model

For completeness, we provide the relevant results for the Gaussian model studied in [85] and [22].

The first two results come in the form of lower-bounds for the minimax detection thresholds under bandwidth- and differential privacy constraints for the distributed signal detection problem presented in the introduction. We recall that in this problem, each local machine $j \in \{1, \ldots, m\}$ observes

$$X_i^{(j)} = f + Z_i^{(j)}, \tag{26}$$

with $f \in \mathbb{R}^d$ and $Z_i^{(j)} \sim N(0, I_d)$, i.i.d. for $i = 1, \ldots, n$. The null hypothesis constitutes that $f = 0$ versus the alternative hypothesis that

$$f \in H_\rho := \left\{ f \in \mathbb{R}^d \ : \ \|f\|_2 \geq \rho \right\}. \tag{27}$$

The following is Theorem 3.1 from [85].

**Theorem 4.** *For each $\alpha \in (0, 1)$ there exists a constant $c_\alpha > 0$ (depending only on $\alpha$) such that if*

$$\rho^2 < c_\alpha \frac{\sqrt{d}}{n} \left( \sqrt{\frac{d}{b \wedge d}} \bigwedge \sqrt{m} \right), \tag{28}$$

*then in the shared randomness protocol case*

$$\inf_{T \in \mathscr{T}_{SR}^{(b)}} \mathcal{R}(H_\rho, T) > \alpha \ \text{for all} \ n, m, d, b \in \mathbb{N}.$$

*Similarly, for*

$$\rho^2 < c_\alpha \frac{\sqrt{d}}{n} \left( \frac{d}{b \wedge d} \bigwedge \sqrt{m} \right), \tag{29}$$

*we have under the local randomness protocol that*

$$\inf_{T \in \mathscr{T}_{LR}^{(b)}} \mathcal{R}(H_\rho, T) > \alpha \ \text{for all} \ n, m, d, b \in \mathbb{N}.$$

Following the proof Theorem 3.1 from [85], we obtain the following lemma.

**Lemma 6.** *Let $\mathscr{T}^b$ denote the class of $b$-bit bandwidth constrained shared- or local randomness distributed testing protocols and let $\rho$ satisfy either (28) or (29), respectively. For any $\alpha \in (0, 1)$, there exists $c_\alpha > 0$ such that for all $T \in \mathscr{T}^{(b)}$ it holds that*

$$\inf_{T \in \mathscr{T}} \mathcal{R}(H_\rho, T) > \alpha - \pi(H_\rho^c),$$

*where $\pi = N(0, c_\alpha^{-1/2} d^{-1} \rho^2 \bar{\Gamma})$ for a symmetric, idempotent matrix $\bar{\Gamma} \in \mathbb{R}^{d \times d}$ with $d/2 \leq rank(\bar{\Gamma}) \leq d$.*

Similarly, the following result can be derived from the proof of Theorem 5 in [22], by taking $s > 0$ in the theorem such that $d_{L_s} = d$.

**Theorem 5.** *For each $\alpha \in (0, 1)$ there exists a constant $c_\alpha > 0$ (depending only on $\alpha$), such that for any $n, m, d \in \mathbb{N}$ and*

$$0 < \epsilon \leq 1 \ and \ 0 \leq \delta \leq \left( c_\alpha m^{-3/2} \wedge nd^{-1}\epsilon^2 \wedge n^{1/2}d^{-1/2}\epsilon^2 \right)^{1+p} \ for \ some \ p > 0, \tag{30}$$

*the condition*

$$\rho^2 < c_\alpha \left( \frac{d}{mn\sqrt{n\epsilon^2 \wedge 1}\sqrt{n\epsilon^2 \wedge d}} \bigwedge \left( \frac{\sqrt{d}}{\sqrt{mn}\sqrt{n\epsilon^2 \wedge 1}} \bigvee \frac{1}{mn^2\epsilon^2} \right) \right), \tag{31}$$

*implies*

$$\inf_{T \in \mathscr{T}_{SR}^{(\epsilon,\delta)}} \mathcal{R}(H_\rho, T) > \alpha.$$

*Similarly, for any $n, m, d \in \mathbb{N}$ and $\epsilon, \delta$ satisfying (30), the condition*

$$\rho^2 < c_\alpha \left( \frac{d\sqrt{d}}{mn(n\epsilon^2 \wedge d)} \wedge \left( \frac{\sqrt{d}}{\sqrt{mn}\sqrt{n\epsilon^2 \wedge 1}} \vee \frac{1}{mn^2\epsilon^2} \right) \right), \tag{32}$$

*implies that*

$$\inf_{T \in \mathscr{T}_{LR}^{(\epsilon,\delta)}} \mathcal{R}(H_\rho, T) > \alpha.$$

Following its proof, we obtain the following the following sub-result.

**Lemma 7.** *Let $\mathscr{T}$ denote the class of shared- or local randomness distributed testing protocols satisfying an $(\epsilon, \delta)$-differential privacy constraint for $0 < \epsilon \leq 1$, $0 \leq \delta \leq \left(c_\alpha m^{-1} \wedge c_\alpha \epsilon m^{-1/2} \wedge n\epsilon^2 \wedge n^2 d^{-1}\epsilon^2 \wedge n^{3/2}d^{-1/2}\epsilon^2\right)$ and let $\rho$ satisfy either (28) or (29), respectively. For any $\alpha \in (0,1)$, there exists $c_\alpha > 0$ such that for all $T \in \mathscr{T}^{(\epsilon,\delta)}$ it holds that*

$$\mathcal{R}(H_\rho, T) > \alpha - \pi(H_\rho^c),$$

*where $\pi = N(0, c_\alpha^{-1/2} d^{-1} \rho^2 \bar{\Gamma})$ for a symmetric, idempotent matrix $\bar{\Gamma} \in \mathbb{R}^{d \times d}$ with $\mathrm{rank}(\bar{\Gamma}) \asymp d$.*

## D    Proofs of Theorems 1, 2 and 3

*Proof of Theorems 1 and 2.* In what follows, let $\mathscr{T}$ denote a class of distributed protocols satisfying either a $b \equiv b_\nu$-bit bandwidth constraint or a local $(\epsilon, \delta)$-differential privacy constraint for $\epsilon \equiv \epsilon_\nu$, $\delta \equiv \delta_\nu$, allowing either for shared randomness or only local randomness.

For any sequences $m \equiv m_\nu$, $d \equiv d_\nu$ and $n \equiv n_\nu$ with $C_R\, md \log d/\sqrt{n} = o(1)$, it follows from Lemma 5 and the bound (14) that the testing risks satisfy

$$\inf_{T \in \mathscr{T}_\mathcal{Q}} \mathcal{R}_{\mathcal{Q}_\nu}(H_{\rho_\nu}, T) = \inf_{T \in \mathscr{T}_\mathcal{P}} \mathcal{R}_{\mathcal{P}_\nu}(H_{\rho_\nu}, T) + o(1). \tag{33}$$

Let $\rho^* \equiv \rho_\nu^*$ be the minimax rate of the $\mathcal{P}$-distributed problem, over the class $\mathscr{T}_\mathcal{P}$, in the sense that $\rho^*$ equals (up to constants) the right-hand side of (6), (7), (9) or (10). We split the proof into showing that $\rho^*$ is an upper and lower-bound for the $\mathcal{Q}$-distributed problem over the class $\mathscr{T}_\mathcal{P}$.

*The rate $\rho^*$ is an upper-bound (up to a poly-logarithmic factor) for the minimax rate in $\mathcal{Q}$:* Write, for $q \in \mathcal{F}$, $\sqrt{q} = (\sqrt{q_i})_{i \in [d]}$. Since $X^{(j)} - \sqrt{q_0}$ is a sufficient statistic for $X^{(j)}$, the model (13) is equivalent in the Le Cam sense the one generated by

$$X^{(j)} = \sqrt{q} - \sqrt{q_0} + \frac{1}{\sqrt{2n}} Z^{(j)} \ \text{ with } Z^{(j)} \sim N(0, I_d), \tag{34}$$

for $q \in \mathcal{F}$, which we shall denote by $\tilde{\mathcal{P}}$. Consequently, by another application of Lemma 5, it suffices to show

$$\inf_{T \in \mathscr{T}_{\tilde{\mathcal{P}}}} \mathcal{R}_{\tilde{\mathcal{P}}}(H_{\rho_\nu}, T) \to 0.$$

If $\|q - q_0\|_1 \geq \rho$, Lemma 15 implies that $\|\sqrt{q} - \sqrt{q_0}\|_2 \geq \rho/2$. Consequently, if $\rho \equiv \rho_\nu \gg M_\nu \rho^*$ where $\rho^*$ is of equal order of the minimax rate for the respective class of distributed protocols $\mathscr{T}_\mathcal{P}$ and $M_\nu$ is an appropriately large factor (of poly-logarithmic order in case of differential privacy constraints), a distributed protocol $T \in \mathscr{T}_\mathcal{P}$ exists for the Gaussian model that achieves the separation rate for whenever $H_0 : \sqrt{q} - \sqrt{q_0} = 0$ versus $H_\rho : \|\sqrt{q} - \sqrt{q_0}\|_2 \geq \rho/2$. By the established equivalence of the minimax risks (33), this implies that a protocol $T \in \mathscr{T}_\mathcal{Q}$ exists for the multinomial model as well. Thus, $\rho_\nu$ is an upper-bound for the minimax separation rate for the class of distributed protocols $\mathscr{T}_\mathcal{Q}$ of the multinomial model.

*The rate $\rho^*$ is a lower-bound for the minimax rate in $\mathcal{Q}$:* Suppose that $\rho \equiv \rho_\nu$ is of smaller order than minimax rate $\rho^*$ of the class $\mathscr{T}_\mathcal{P}$, in the sense that $\rho^*/\rho \to \infty$ as $\nu \to \infty$. We aim to use the Bayes risk lower-bound of Lemmas 6 and 7, which apply to a Gaussian prior. To accommodate

a Gaussian prior with sufficient mass on the alternative hypothesis, we first need to address the "constraint on the signal" imposed by $\sum_{i=1}^{d} q_i = 1$ for $q \in \mathcal{F}$.

To that extent, consider without loss of generality $d$ to be divisible by two. Let $I_R := [-(R-1)/(R+1), (R-1)/(R+1)]$. For all $(f_i)_{i \in [d/2]} \in I_R^{d/2}/\sqrt{d}$, there exists a $q^f := (q_i^f)_{i \in [d]} \in \mathcal{F}$ such that $q_i^f = 1/d + f_i/\sqrt{d}$ for $i = 1, \ldots, d/2$ and $q_i^f = 1/d - f_i/\sqrt{d}$ for $i = d/2+1, \ldots, d$. To see that $q^f \in \mathcal{F}$, note that $\sum_{i=1}^{d} q_i^f = 1$, $q^f \geq 0$ and

$$\max_{1 \leq i,k \leq d} \frac{q_i^f}{q_k^f} \leq \max_{c \in I_R} \frac{1+c}{1-c} = R.$$

Define $\mathcal{F}'$ as the set

$$\left\{ (q_i)_{i \in [d]} \in \mathcal{F} \; : \; (f_i)_{i \in [d/2]} \in \frac{I_R^{d/2}}{\sqrt{d}} \text{ s.t. } q_i^f = 1/d + (1 - 2\mathbb{1}_{i>d/2})\frac{f_i}{\sqrt{d}} \text{ for } i = 1, \ldots, d \right\}$$

and

$$H_\rho' := \left\{ q \; : \; q \in \mathcal{F}', \|q - q_0\|_1 \geq \rho \right\}.$$

We have $\mathcal{F}' \subset \mathcal{F}$, which in turn implies that $H_\rho' \subset H_\rho$. Combined with the fact that the testing risk decreases by considering smaller alternative hypotheses, this results in

$$\inf_{T \in \mathscr{T}_\mathcal{P}} \mathcal{R}_\mathcal{P}(T, H_\rho) \geq \inf_{T \in \mathscr{T}_\mathcal{P}} \mathcal{R}_\mathcal{P}(T, H_\rho'). \tag{35}$$

Define $g_f = (1/2)(f, -f) \in \mathbb{R}^d$. By Pinsker's inequality,

$$\left\| P_{\sqrt{q^f} - \sqrt{q_0}}^{nm} - P_{g_f}^{nm} \right\|_{\mathrm{TV}} \leq 1 \wedge \sqrt{\frac{mn}{4} D_{\mathrm{KL}}(P_{\sqrt{q} - \sqrt{q_0}}; P_{g_f})}$$

$$= 1 \wedge \frac{\sqrt{mn}}{2} \left\| \sqrt{q_0 + 2g_f/\sqrt{d}} - \sqrt{q_0} - g_f \right\|_2 =: D_f,$$

where $P_{\sqrt{q} - \sqrt{q_0}}^n$ denotes the distribution of (34) and the square root is to be understood as applied coordinate wise.

Let $\pi = N(0, d^{-1}(\rho^*)^2 \bar{\Gamma})$ for a symmetric, idempotent matrix $\bar{\Gamma} \in \mathbb{R}^{d/2 \times d/2}$ with $d/4 \leq \mathrm{rank}(\bar{\Gamma}) \leq d/2$.

We have that

$$\inf_{T \in \mathscr{T}_\mathcal{P}} \mathcal{R}_\mathcal{P}(T, H_\rho') \geq \inf_{T \in \mathscr{J}_\mathcal{P}} \left[ \mathbb{P}_0 T(Y) + \int \mathbb{P}_{g_f}(1 - T(Y)) d\pi(f) \right] - 2 \int D_f d\pi(f)$$

$$- \pi \left( f : f \notin (I_R/\sqrt{d})^{d/2} \text{ or } \left\| (q_i^f)_{i \in [d]} - q_0 \right\|_1 < \rho \right).$$

By Lemma 8, the model $\{P_{g_f} \; : \; f \in I_R^{d/2}/\sqrt{d}\}$ is equivalent to the model generated by the observations

$$S_i^{(j)} := f_i + \frac{1}{\sqrt{n}} Z_i^{(j)} \tag{36}$$

for $i = 1, \ldots, d/2$. Since $\mathcal{F}'$ is bijective with $(I_R/\sqrt{d})^{d/2}$, Lemma 5 implies that

$$\inf_{T \in \mathscr{J}_\mathcal{P}} \left[ \mathbb{P}_0 T(Y) + \int \mathbb{P}_{g_f}(1 - T(Y)) d\pi(f) \right] = \inf_{T \in \mathscr{J}_{\tilde{\mathcal{P}}}} \left[ \mathbb{P}_0' T(Y) + \int \mathbb{P}_f'(1 - T(Y)) d\pi(f) \right] \tag{37}$$

where $\tilde{\mathcal{P}}$ is the model generated by the observations in display (36) for $i = 1, \ldots, d/2$ and $\mathbb{P}_f'$ denotes the distribution of the distributed protocol with data generated from $f \in \tilde{\mathcal{P}}$.

It follows from Lemma 6 in the case of bandwidth constraints or Lemma 7 in the case of privacy constraints (using that $\rho \ll \rho^*$ in both cases) that the latter distributed testing risk is lower-bounded by

$$1 - o(1) - \pi \left( f \in \mathbb{R}^{d/2} \; : \; f \notin (I_R/\sqrt{d})^{d/2} \text{ or } \left\| (q_i^f)_{i \in [d]} - q_0 \right\|_1 < \rho \right) - 2 \int D_f d\pi(f). \tag{38}$$

Addressing the third term in the display above; the theorem(s) assume that $md\log d/\sqrt{n} \overset{\nu\to\infty}{\to} 0$, $b \geq 1$ and $\epsilon \gg n^{-1/4}$, we have that $\rho^* \ll 1/\sqrt{\log(d)}$, which gives $\|\sqrt{d}f_i\|_\infty \to 0$ with $\pi$-probability tending to one (see e.g. Lemma 16), which in turn implies that

$$f_i \in I_R/\sqrt{d} \quad \text{for all } i = 1, \ldots, d/2. \tag{39}$$

Next, we show that $\|(q_i^f)_{i\in[d]} - q_0\|_1 \geq \rho$ with $\pi$-probability tending to one. Since $\sum_{i=1}^d |q_i^f - q_0| = 2\sum_{i=1}^{d/2} |f_i/\sqrt{d}|$, we have that for some constants $c, c' > 0$,

$$\pi\left(\|q^f - q_0\|_1 < \rho\right) \leq \pi\left(\left\|f/\sqrt{d}\right\|_1 < \rho\right) \leq 1 - \Pr\left(\|\bar{\Gamma}Z\|_1 \geq c'd\frac{\rho}{\rho^*}\right).$$

where in the expression on the right-hand side, $Z \sim N(0, I_{d/2})$. Since $\rho \ll \rho^*$ and $\bar{\Gamma}$ is idempotent with rank of the order $d$, we can conclude that the expression vanishes. This takes care of the third term in (38).

For the last term in (38), the Taylor approximation $\sqrt{1+y} - 1 = y/2 - y^2/8 + \frac{y^3}{16(1+\eta_y^{5/2})}$ for some $\eta \in [0, y]$, combined with the fact that $\|\sqrt{d}f\|_\infty = o_\pi(1)$ yields that

$$2\int D_f d\pi(f) \leq \int 1 \wedge \sqrt{mnd} \left\|(f_i^2)_{i\in[d/2]}\right\|_2 d\pi(f) \lesssim \sqrt{mn}\rho^2.$$

Since the theorem(s) assume that $md\log d/\sqrt{n} \overset{\nu\to\infty}{\to} 0$, $b \geq 1$ and $\epsilon \gg n^{-1/4}$, the right-hand side of the above display vanishes.

$\square$

*Proof of Theorem 3.* Let $\mathscr{T}_Q$, $\mathscr{T}_P$ denote the class of distributed $b$-bit bandwidth constrained testing protocols with $b \in \mathbb{N}$, $m \in \mathbb{N}$, $d \in 2\mathbb{N}$ and $n \in \mathbb{N}$ and no access to shared randomness for the models $Q$ and $P$, respectively. We note here that under the conditions of the theorem, we can assume $d$ and $n$ are both larger than some constant; and in particular we can assume $d \in 2\mathbb{N}$ without loss of generality. Assume $d$ and $n$ satisfy (15), for a constant $C$ to be set later. The proof follows by the fact that the distributed testing problems have different minimax testing rates, for certain values of $b$ and $m$.

Consider the hypothesis test given in (4), with $H_0 : q_0 = (1/d, \ldots, 1/d) \in \mathbb{S}^d$ and $H_\rho$ as in the display.

Set $b = \lceil n\log_2(d) \rceil$. When $b \geq n\log_2(d)$, the observations $\tilde{X}^{(j)}$ in the multinomial model as given in (3) are valid $b$-bit transcripts, since $|\{1, \ldots, d\}^n| \leq n\log_2(d)$. These transcripts are therefore sufficient for the nondistributed / unconstrained model $Q^m$, i.e. corresponding to observations

$$\tilde{X} \sim Q_{q,nm} \quad \text{for } q \in \mathcal{F}.$$

Consequently, the distributed, $b$-bit bandwidth constraint testing risk for $Q$ is equal to the testing risk $Q^m$;

$$\inf_{T\in\mathscr{T}_Q} \mathcal{R}_Q(H_\rho, T) = \inf_T \mathcal{R}_{Q^m}(H_\rho, T).$$

This means that, for all $\alpha \in (0, 1)$, there exists $C_\alpha > 0$ and a distributed protocol $T$ satisfying a $b$-bandwidth constraint for distributed experiment $Q$ such that

$$\inf_{T\in\mathscr{T}_Q} \mathcal{R}_Q(H_\rho, T) < \alpha \text{ whenever } \rho^2 \geq C_\alpha \frac{\sqrt{d}}{mn}$$

where $H_\rho$ as defined in (4), as the minimax rate for the unconstrained problem with $mn$ observations is $\rho_{Q^m}^2 := \sqrt{d}/(mn)$ (see e.g. Theorem 3 in [70]).

On the other hand, whenever $mb = m\lceil n\log_2(d)\rceil \leq d$, the minimax rate for the distributed testing risk of $P$ for the (comparable) hypotheses

$$H_0 : q = q_0 \quad \text{versus} \quad \tilde{H}_\rho : \|\sqrt{q} - \sqrt{q_0}\|_2 \geq \rho$$

is bounded from below by $\rho_P^2 \asymp \sqrt{d}/(\sqrt{mn})$, as a consequence of Theorem 4. Specifically, following the proof of Theorem 1 above, we have that

$$\inf_{T\in\mathscr{T}_P} \mathcal{R}_P(H_\rho, T) \geq \inf_{T\in\mathscr{T}_{\tilde{P}}} \mathcal{R}_{\tilde{P}}(\tilde{H}_\rho, T),$$

where

$$\tilde{H}_\rho := \left\{ f \in (I_R/\sqrt{d})^{d/2} : \|f\|_1 \geq \rho \right\},$$

for $I_R := [\sqrt{2}(1-\sqrt{R})/\sqrt{1+R}, \sqrt{2}(\sqrt{R}-1)/\sqrt{1+R}]$, $\tilde{\mathcal{P}}$ is generated by the observations

$$X^{(j)} = f + \frac{1}{\sqrt{n}} Z^{(j)}$$

for $Z^{(j)} \sim N(0, I_{d/2})$, indexed by $f \in (I_R/\sqrt{d})^{d/2}$ and the class $\mathscr{T}_{\tilde{\mathcal{P}}}$ is to be understood as the $b$-bit bandwidth constraint distributed testing protocols for the model $\tilde{\mathcal{P}}$ and $j = 1, \ldots, m$ machines.

Lemma (6) implies that for all $\alpha \in (0, 1)$ the latter is bounded by

$$\alpha - N(0, c_\alpha^{-1/2} d^{-1} \rho^2 \bar{\Gamma}) \left( \tilde{H}_\rho^c \right),$$

for a symmetric, idempotent matrix $\bar{\Gamma} \in \mathbb{R}^{d/2 \times d/2}$ with $d/4 \leq \mathrm{rank}(\bar{\Gamma}) \leq d/2$ $\alpha \in (0, 1)$, whenever $\rho^2 \leq c_\alpha \frac{\sqrt{d}}{\sqrt{mn}}$ for some small enough constant $c_\alpha > 0$. By the same analysis as conducted in the proof of Theorem 1 above (using that $n \leq \log(d)$), we find that the second term is at most $\alpha/2$ for $c_\alpha > 0$ small enough. Summarizing, we find in particular that for some constant $c_\alpha > 0$,

$$\inf_{T \in \mathscr{T}_{\mathcal{P}}} \mathcal{R}_{\mathcal{P}}(T, H_\rho) > 1/3,$$

for all $\rho^2 \leq c\sqrt{d}/(\sqrt{m}n)$ and $m, n, b, d$ such that $mb \leq d$, where the number $1/3$ is chosen without particular significance.

Whenever $mb = m\lceil n \log_2(d) \rceil \leq d$,

$$\inf_{T \in \mathscr{T}_{\mathcal{Q}}} \mathcal{R}_{\mathcal{Q}}(H_\rho, T) < 1/6 < 1/3 < \inf_{T \in \mathscr{T}_{\mathcal{P}}} \mathcal{R}_{\mathcal{P}}(H_\rho, T). \tag{40}$$

for some $C_\alpha > 0$ large enough and $c_\alpha > 0$ small enough. Take the constant $C = \lceil C_\alpha^2/c_\alpha^2 \rceil$ such that if $m = C$, it holds that

$$C_\alpha \frac{\sqrt{d}}{mn} \leq \rho^2 \leq c_\alpha \frac{\sqrt{d}}{\sqrt{mn}}, \quad \text{with } \rho^2 := C_\alpha \frac{\sqrt{d}}{\sqrt{M}\sqrt{mn}}.$$

Now suppose that $C\mathfrak{d}(\mathcal{Q}, \mathcal{P}) \leq 1/6$. Lemma 5 then implies in that

$$\inf_{T \in \mathscr{T}_{\mathcal{P}}} \mathcal{R}_{\mathcal{P}}(H_\rho, T) \leq \inf_{T \in \mathscr{T}_{\mathcal{Q}}} \mathcal{R}_{\mathcal{Q}}(H_\rho, T) + 1/6 < 1/3.$$

This contradicts (40). We conclude that

$$C\mathfrak{d}(\mathcal{Q}, \mathcal{P}) > 1/6, \tag{41}$$

whenever $d/\lceil n \log_2(d) \rceil > C$. The result now follows with $c = 1/(6C)$. $\qquad \square$

## E   Auxilliary lemmas

The following lemma is used in the comparison of the multinomial model to the many-normal-means model.

**Lemma 8.** *Let $d \in 2\mathbb{N}$, $\mathcal{F} \subset \mathbb{R}^{d/2}$, and consider for $i = 1, \ldots, d$ independent random variables $X_i = h_i + \sigma Z_i$ with $\sigma > 0$ and $Z_i \sim N(0, 1)$ satisfying*

$$h_i = \begin{cases} a_i f_i & \text{if } i \leq d/2, \\ -a_i f_{i-d/2} & \text{if } i > d/2, \end{cases}$$

*for some $f \in \mathcal{F}$ and $a = (a_i)_{i \in [d]} \in \mathbb{R}^d$. Let $\mathcal{P}$ denote the model generated by the observations $X := (X_1, \ldots, X_d) \sim P_f$, $f \in \mathcal{F}$ and let $\mathcal{Q}$ denote the model generated by*

$$\tilde{X}_i = (a_i + a_{d/2+i}) f_i + \sqrt{2} \sigma Z_i, \quad \text{for } i = 1, \ldots, d/2,$$

*with $Z_i \overset{i.i.d.}{\sim} N(0, 1)$ and $f \in \mathcal{F}$.*

*Then, $\Delta(\mathcal{P}, \mathcal{Q}) = 0$.*

*Proof.* The statistic $S = (a_i X_i - a_i X_{d/2+1})_{i \in [d]}$ is sufficient for the model $\mathcal{P}$ by using Neyman-Fisher (Lemma 2). We have

$$\frac{dP_f}{dP_0}(X) = \prod_{i=1}^{d} \exp\left(\sigma^{-1} X_i h_i - \frac{1}{2\sigma^2} h_i^2\right)$$

$$= \prod_{i=1}^{d/2} \exp\left(\sigma^{-1}(a_i X_i - a_i X_{d/2+1}) f_i - \frac{1}{\sigma^2} f_i^2\right) \qquad = e^{\sigma^{-1} S^\top f - \frac{1}{\sigma^2} \|f\|_2^2}.$$

In distribution, $\tilde{X} = (\tilde{X}_i)_{i \in [d]}$ is equal to $S$, which implies $\Delta(\mathcal{P}, \mathcal{Q}) = 0$ per Lemma 2. $\qquad\square$

The following lemmas are well known but included for completeness.

**Lemma 9.** *Let $P_f$ denote the distribution of a $N(f, \sigma I_d)$ distributed random vector for $f \in \mathbb{R}^d$ and let $P_f^n$ denote the distribution of $n$ i.i.d. draws (i.e. $P_f^n = \bigotimes_{i=1}^n P_f$).*

*It holds that*

$$\left\| P_f^n - P_g^n \right\|_{\mathrm{TV}} \leq \frac{n}{2\sigma} \|f - g\|_2.$$

*Proof.* By Pinsker's inequality,

$$\|P_f^n - P_g^n\|_{\mathrm{TV}} \leq \sqrt{\frac{n}{2} D_{\mathrm{KL}}(P_f; P_g)}.$$

A straightforward calculation gives that the latter is bounded by $\frac{\sqrt{n}}{2\sigma} \|f - g\|_2$. $\qquad\square$

The following lemma relates the total variation distance between $P, Q$ to the $L_1$-distance between corresponding densities.

**Lemma 10.** *Let $P, Q$ be probability measures dominated by a sigma-finite measure $\mu$ with corresponding probability densities $p = \frac{dP}{d\mu}$ and $q = \frac{dQ}{d\mu}$. It holds that*

$$\|P - Q\|_{\mathrm{TV}} = \frac{1}{2} \int |p(x) - q(x)| d\mu(x).$$

*Proof.* See e.g. Section 2.4 in [93]. $\qquad\square$

**Lemma 11.** *For any two probability measures $P$ and $Q$ on a measurable space $(\mathcal{X}, \mathcal{X})$ with $\mathcal{X}$ a Polish space and $\mathcal{X}$ its Borel sigma-algebra. There exists a coupling $\mathbb{P}^{X, \tilde{X}}$ such that*

$$\|P - Q\|_{\mathrm{TV}} = 2\mathbb{P}^{X, \tilde{X}}\left(X \neq \tilde{X}\right).$$

*Proof.* See e.g. Section 8.3 in [91]. $\qquad\square$

The next lemma gives a useful characterization of the total variation distance between two probability measures.

**Lemma 12.** *Let $P$ be a signed, bounded measure defined on measurable space $(\mathcal{X}, \mathcal{X})$ and suppose that $P \ll \nu$ for a sigma-finite measure $\nu$. It holds that*

$$\|P\|_{\mathrm{TV}} = \frac{1}{2} \sup\left\{ \int f dP : |f| \leq 1 \text{ and } f : \mathcal{X} \to \mathbb{R} \text{ is measurable} \right\}. \tag{42}$$

*Proof.* Consider the Jordan measure decomposition $P = P^+ - P^-$, where $P^+, P^-$ are both positive, bounded measures such that $P^+ \perp P^-$. For any measurable $f$, $\{f \geq 0\}, \{f \leq 0\} \in \mathcal{X}$, so $|f| \leq 1$ means that

$$\int f dP \leq \int f \mathbb{1}_{\{f \geq 0\}} dP^+ - \int f \mathbb{1}_{\{f \leq 0\}} dP^-$$

$$\leq \int \mathbb{1}_{\{f \geq 0\}} dP^+ + \int \mathbb{1}_{\{f \leq 0\}} dP^-$$

$$\leq \|P^+\|_{\mathrm{TV}} + \|P^-\|_{\mathrm{TV}} \leq 2\|P\|_{\mathrm{TV}}.$$

For the other direction, note that $f = \text{sign}(p - q)$ is measurable and bounded by 1, which gives

$$\frac{1}{2} \int f dP = \int |p - q| d\nu = \|P - Q\|_{\text{TV}},$$

where the last equality follows from Lemma 10. □

**Lemma 13.** *Let $P = \bigotimes_{j=1}^{m} P_j$ and $Q = \bigotimes_{j=1}^{m} Q_j$ for probability measures $P_j, Q_j$ defined on a common measurable space $(\mathcal{X}, \mathscr{X})$, with probability densities $p_j, q_j$ for $j = 1, \ldots m$. It holds that*

$$\|P - Q\|_{\text{TV}} \le \sum_{j=1}^{m} \|P_j - Q_j\|_{\text{TV}}.$$

*Proof.* The measures $P_j$ and $Q_j$ admit densities with respect to $P_j + Q_j$, which we shall denote by $p_j$ and $q_j$, respectively, with

$$p := \prod_{j=1}^{m} p_j = \frac{d \bigotimes_{j=1}^{m} P_j}{d \bigotimes_{j=1}^{m}(P_j + Q_j)} \quad \text{and} \quad q := \prod_{j=1}^{m} q_j = \frac{d \bigotimes_{j=1}^{m} Q_j}{d \bigotimes_{j=1}^{m}(P_j + Q_j)}.$$

Writing $\mu = \bigotimes_{j=1}^{m}(P_j + Q_j)$ and applying Lemma 10 we obtain

$$\|P - Q\|_{\text{TV}} = \frac{1}{2} \int |\prod_{j=1}^{m} p_j(x_j) - \prod_{j=1}^{m} q_j(x_j)| d\mu(x_1, \ldots, x_m). \tag{43}$$

By the telescoping product identity

$$a_1 \cdot a_2 \cdots a_m - b_1 \cdot b_2 \cdots b_m = \sum_{j=1}^{m}(a_j - b_j) \prod_{k=1}^{j-1} a_k \prod_{k=j+1}^{m} b_k \tag{44}$$

and Fubini's Theorem, the right-hand side of (43) is bounded by

$$\sum_{j=1}^{m} \frac{1}{2} \int |p_j(x_j) - q_j(x_j)| d(P_j + Q_j)(x_j) = \sum_{j=1}^{m} \|P_j - Q_j\|_{\text{TV}}.$$

□

The following lemma can be seen as a data processing inequality for the total variation distance.

**Lemma 14.** *Let $(\mathcal{X}, \mathscr{X})$ and $(\mathcal{Y}, \mathscr{Y})$ be two measurable spaces and let $K : \mathcal{Y} \times \mathscr{X} \to [0, 1]$ be a Markov kernel. For any probability measures $P, Q$ defined on $\mathscr{X}$ it holds that*

$$\|PK - QK\|_{\text{TV}} \le \|P - Q\|_{\text{TV}}.$$

*Proof.* This follows immediately from the representation in Lemma 12 combined with the fact that, for $|f| \le 1$, $x \mapsto \int f(y) dK(y|x)$ is a measurable function bounded by 1, since $K$ is Markov kernel. Hence,

$$\sup_{A} |PK(A) - QK(A)| = \frac{1}{2} \sup_{f} \int \int f(y) dK(y|x) d(P - Q)(x)$$

$$\le \frac{1}{2} \sup_{f} \int f(x) d(P - Q)(x).$$

□

The next lemma bounds the $L_1$-distance $\|p - q\|_1$ between densities with a multiple of the Hellinger distance $2^{-1/2} \|\sqrt{p} - \sqrt{q}\|_2$.

**Lemma 15.** *For two probability densities $p, q$ with respect to $\mu$, it holds that*

$$\frac{1}{2} \int |p(x) - q(x)| d\mu(x) \le \sqrt{\int \left(\sqrt{p(x)} - \sqrt{q(x)}\right)^2 d\mu(x)}.$$

*Proof.* The result follow from the Cauchy-Schwarz inequality and the fact that $\int p d\mu = \int q d\mu = 1$. See e.g. [93] for details. $\square$

**Lemma 16.** *Let $K \in \mathbb{N}$ and $M \in \mathbb{R}^{K \times K}$ be symmetric and positive definite. Consider the random vector $G = (G_1, \ldots, G_K) \sim N(0, M)$. It holds that $\mathbb{E} \max_{1 \leq i \leq K} |G_i| \leq 3\|M\| \sqrt{\log(K) \vee \log(2)}$ and*

$$Pr\left( \max_{1 \leq i \leq K} G_i^2 \geq \|M\|^2 x \right) \leq \frac{2K}{e^{x/4}},$$

*for all $x > 0$.*

*Proof.* It holds that

$$G \overset{d}{=} \sqrt{M} Z, \quad \text{with} \quad Z \sim N(0, I_K).$$

Since $M$ is symmetric, positive definite, it has SVD decomposition $M = V \mathrm{Diag}(\lambda_1, \ldots, \lambda_K) V^\top$. Since $V$ is orthonormal,

$$\sqrt{M} Z = V \sqrt{\mathrm{Diag}(\lambda_1, \ldots, \lambda_K)}(V^\top Z) \overset{d}{=} V \sqrt{\mathrm{Diag}(\lambda_1, \ldots, \lambda_K)} Z.$$

Writing $V = [v_1 \ \ldots \ v_K]$ where $v_k$ are orthogonal unit vectors, the latter display equals

$$\sum_{k=1}^{K} \sqrt{\lambda_k} v_k Z_k \ \sim \ N\left(0, \mathrm{Diag}(\lambda_1, \ldots, \lambda_K)\right).$$

Consequently,

$$\max_{k \in [K]} |G_k| \overset{d}{=} \max_{k \in [K]} |\lambda_k Z_k| \leq \|M\| \max_{k \in [K]} |Z_k|.$$

Hence, it suffices to show that

$$\Pr\left( \max_{1 \leq i \leq K} Z_i^2 \geq x \right) \leq \frac{2K}{e^{x/4}}.$$

The case where $K = 1$ follows by standard Gaussian concentration properties. Assume $K \geq 2$. For $0 \leq t \leq 1/4$,

$$\mathbb{E} e^{t \max_i (Z_i)^2} = e^t \mathbb{E} \max_i e^{t(Z_i^2 - 1)} \leq K e^{2t^2 + t}.$$

Taking $t = 1/4$ and applying Markov's inequality yields the second statement of the lemma. Furthermore, in view of Jensen's inequality

$$\mathbb{E} \max_i (Z_i)^2 \leq \frac{\log(K)}{t} + 2t + 1,$$

which in turn yields $\mathbb{E} \max_i |Z_i| \leq 3\sqrt{\log(K)}$. $\square$

