# OpenReview forum: "Optimal Private and Communication Constraint Distributed Goodness-of-Fit Testing for Discrete Distributions in the Large Sample Regime"
_NeurIPS.cc/2024/Conference — NeurIPS 2024 poster_

### Official Review · Reviewer_3Xr9 · 2024-07-10

**Soundness:** 3
**Presentation:** 3
**Contribution:** 2
**Rating:** 6
**Confidence:** 3

**Summary:**

The paper focuses on the minmax rate for goodness-of-fit testing for discrete distributions under bandwidth and differential privacy constraints in a distributed setting, leveraging Le Cam’s theorem. The main distinction from previous literature lies in the consideration of the distributed setting.

**Strengths:**

The paper is well-written, well-organized, and mathematically rigorous, with a clear exposition of the concepts using probability theory. The key contribution is the extension of the optimal rate for goodness-of-fit testing under differential privacy to the distributed case. However, I have concerns regarding this extension, which I detail in Weaknesses.

**Weaknesses:**

I might be wrong, but I didn't see the difference between the distributed formulation in this paper and the central setting in other works (where data is available at one central location). The only exception is that the distributed formulation posits that each server adopts a local protocol (local privacy-mapping with bandwidth constraints). However, given that the raw data
$X^{(j)},j=1,\ldots,m$ are i.i.d., wouldn’t the optimal rate be achieved when all local protocols are equivalent? If one server has a better protocol than others, the risk defined after line 190 is not minimized. If this is true, the question is what distinguishes the proposed distributed protocol from the privacy-preserving case in a central setting? For the latter, the min-max rate has already been derived, as seen in:

"Local Privacy and Statistical Minimax Rates"

"Robust Estimation of Discrete Distributions under Local Differential Privacy"

"The Cost of Privacy: Optimal Rates of Convergence Performance Estimation with Differential Privacy"

The paper needs to clearly articulate the differences and potential advantages of the distributed approach over these established results in the central setting. Without this clarification, the novelty and implications of the results may be unclear.

My score will change based on the authors' response to this question.

**Questions:**

See Weaknesses

**Limitations:**

The authors adequately addressed the limitations.

---

> ### Author Rebuttal · Authors · 2024-08-07
>
> We sincerely appreciate the time and effort dedicated to evaluating our paper and we thank you for your thoughtful review. We appreciate your recognition of the paper’s mathematical rigor, organization, and clarity. We would like to address the concerns and the question you raised regarding the differences between our distributed setting and the central setting in existing literature.
>
> "*The paper needs to clearly articulate the differences and potential advantages of the distributed approach over these established results in the central setting. Without this clarification, the novelty and implications of the results may be unclear.*"
>
> Thank you for this constructive feedback. In our revised version of the manuscript, we now describe this difference explicitly, see also our response to your question below.
>
> "*I might be wrong, but I didn't see the difference between the distributed formulation in this paper and the central setting in other works (where data is available at one central location). The only exception is that the distributed formulation posits that each server adopts a local protocol (local privacy-mapping with bandwidth constraints). However, given that the raw data are i.i.d., wouldn’t the optimal rate be achieved when all local protocols are equivalent? If one server has a better protocol than others, the risk defined after line 190 is not minimized. For the latter, the min-max rate has already been derived, as seen in: [15,16,17]*"
>
> This is a natural question to ask. The testing risk, as defined after line 190, is non-linear in the transcripts $Y = (Y^{(1)},\ldots,Y^{(m)})$, (i.e. the output of the servers). As a result, it is not apriori clear whether optimal federated procedures have each server follow the same locally optimal protocol, even though the data is i.i.d. for each of the servers. It turns out that for certain values of $n$, $m$, $d$ and `contraint-budget' $b$ or $\epsilon$, optimal distributed protocols have each server execute a different strategy. For example, for small budgets $b$, optimal certain servers might communicate information which only concerns a specific part of a partition of the domain, with each server "covering" a different part of the partition. We refer for the details of these methods of Section 4 in [11] and Section 3 in [12], whose methods extend to the discrete distribution setting using the theory developed in our paper.
>
> What separates the federated setting considered by us and e.g. [1,2,3,4,5,6,7,11,12] from [15,16,17] is indeed that protocol adopted by each server satisfies an individual bandwidth or privacy constraint, instead of just the final output of the inference (i.e. the final estimator in [15,16,17]). This turns out to have a profound impact on the best possible theoretical performance. Due to the non-linear nature of the risk considered, the differences between these settings are substantial, both in terms of optimal methods and best possible performance. Our theory underlines this general finding. For example, the rate described by Theorem 1 yields that there is always a benefit of increasing the number of servers $m$, as the separation rate decreases as $1/\sqrt{m}$ (or faster, depending on $b$ and $d$). The non-linearity of the risk is what drives this difference between the two types of settings. If instead one would consider e.g. the average of local risks, the results would reflect those in the central / single server setting. In certain estimation settings, such an "average of local losses" makes sense to consider in a federated framework. For the problem of goodness-of-fit testing we consider, however, the non-linear nature of the problem is inherent. We have revised our explanation in the article to better reflect these differences.

---

> > ### Comment · Reviewer_3Xr9 · 2024-08-14
> >
> > Thanks for your response. My questions have been addressed. I'll raise the score.

---

### Official Review · Reviewer_4qfs · 2024-07-12

**Soundness:** 3
**Presentation:** 3
**Contribution:** 2
**Rating:** 6
**Confidence:** 2

**Summary:**

This paper explores distributed goodness-of-fit testing for discrete distributions under bandwidth and differential privacy constraints. The authors extend results from multivariate Gaussian models using Le Cam’s theory of statistical equivalence. They derive matching minimax upper and lower bounds for the goodness-of-fit testing problem when the number of samples held locally is large.

**Strengths:**

The framework presented for extending goodness-of-fit testing from Gaussian models to discrete distributions is novel and addresses practical issues in federated learning scenarios. The derivation of matching minimax upper and lower bounds is rigorous and thorough, leveraging statistical equivalence effectively. The paper addresses key challenges in distributed settings, specifically under bandwidth and privacy constraints, which are crucial for modern applications in federated learning.

**Weaknesses:**

The results rely on the assumption $md\log d/\sqrt{n}=o(1)$, which can be attained when the number of data is large. When $n$ is large, the setting naturally gets close to the Gaussian case, from which some existing tools can be leveraged. In this sense, the analyses presented in this paper are not too surprising. Moreover, the absence of empirical validation or simulations to demonstrate the practical performance of the theoretical results limits the impact of the findings in practice.

**Questions:**

- How practical are the assumption $md\log d/\sqrt{n}=o(1)$ in real-world federated learning scenarios? Can you provide examples or case studies where these conditions hold?

- The paper assumes large sample regimes for the derivation of the minimax rates. How would the results change if the sample size was not large? Are there any extensions or modifications of the theory to handle smaller sample sizes?

- How does the proposed method compare with existing methods for distributed goodness-of-fit testing in terms of computational efficiency and communication overhead?

**Limitations:**

The authors adequately addressed the limitations.

---

> ### Author Rebuttal · Authors · 2024-08-07
>
> Thank you for the time and effort invested in evaluating our work, the kind words, the constructive feedback and the interesting questions raised. We respond pointwise below.
>
> *"The paper addresses key challenges in distributed settings, specifically under bandwidth and privacy constraints, which are crucial for modern applications in federated learning."*
>
> Thank you for highlighting the importance of our work.
>
> *"The results rely on the assumption $m d \log d / \sqrt{n} = o(1)$, which can be attained when the number of data is large. Moreover, the absence of empirical validation or simulations to demonstrate the practical performance of the theoretical results limits the impact of the findings in practice."*
>
> The problem of federated goodness-of-fit testing for discrete distributions under bandwidth and differential privacy constraints is notoriously difficult to study (see also our reply to Reviewer (Nun8)). Just like the works covering the $n=1$ case, this work poses just one step in uncovering the principle phenomena underlying the problem. Whilst it is true that our work is theoretical in nature, and covers only the large local data regime, we believe the results offer practical guidance as well. For example, it provides guidance on whether methods that work for Gaussian models also perform well in the discrete setting, as turns out to be the case in the large local data regime. Furthermore, the optimal performance described by the separation rate provides a benchmark for further development of both methods and theory.
>
> *"How practical are the assumption $m d \log d / \sqrt{n} = o(1)$ in real-world federated learning scenarios? Can you provide examples or case studies where these conditions hold?"*
>
> There are interesting real-world federated scenarios where $m d \log d / \sqrt{n} = o(1)$ is feasible. One such example is text mining, where even though the dictionary $d$ is large, it is feasible that the number of words mined $n$ is much larger still. Scenarios where $d$ is small to begin with might not be so interesting under bandwidth constraints, but often still are when considering differential privacy; for example in the case of categorical data observed by $m$ different hospitals adhering to a privacy constraint, with a patient pool of $n \gg m^2$ in each hospital.
>
> *"The paper assumes large sample regimes for the derivation of the minimax rates. How would the results change if the sample size was not large? Are there any extensions or modifications of the theory to handle smaller sample sizes?"*
>
> For goodness-of-fit testing in the Gaussian model, non-asymptotic minimax results are available (i.e. holding for any $d,n,m \in \mathbb{N}$), derived for bandwidth constraints in [11] and differential privacy constraints in [12]. For goodness-of-fit testing in discrete distributions under bandwidth and differential privacy constraints, the results of [13] assume $n=1$, but extend to any $n \asymp 1$ without a change in the minimax rate. Matching upper- and lower-bounds have not been derived in the literature in the intermediate regime where $m^2 d^2 \log^2 d \lesssim n \gg 1$, however. We have tried, but it turns out that goodness-of-fit testing for discrete distributions is more difficult in the federated setting. The techniques of [11,12] seem very specialized for Gaussian models, and the essentially combinatorial argument of [13] becomes cumbersome for large $n$, see also our reply to Reviewer (Nun8).
>
> *"How does the proposed method compare with existing methods for distributed goodness-of-fit testing in terms of computational efficiency and communication overhead?"*
>
> This is an interesting question with a slightly involved answer. The overhead of transforming the discrete observations to "Gaussian-like" data as proposed in for example [14] is small; it essentially consists adding uniform perturbations to average frequencies. The overhead of the rate-optimal procedures in the Gaussian models depends heavily on whether bandwidth or differential privacy constraints are under consideration. Under bandwidth constraints, the methods proposed for Gaussian data in [11] enjoy decent computational overhead, depending on the "regime" (i.e. large $b$ compared to $m$ and $d$). We refer to Section 4 in [11] for details. Under differential privacy constraints, existing rate-optimal methods in the Gaussian models, for example those used in [12], are computationally expensive in $d$ in the regime where $\epsilon \ll \sqrt{d}/\sqrt{m}$ for shared randomness protocols, and $\epsilon \ll d / \sqrt{m}$ for local randomness protocols. Computationally more feasible methods exist, but those attain slightly worse rates. We refer to [9, 10] for a discussion. In future research, we hope to develop methods that computationally efficient and rate optimal for the goodness-of-fit testing problem with discrete distributions. We included comments on this in the revised version of our manuscript.

---

### Official Review · Reviewer_aGfz · 2024-07-12

**Soundness:** 4
**Presentation:** 4
**Contribution:** 2
**Rating:** 7
**Confidence:** 3

**Summary:**

This paper investigates the problem of Goodness-of-fit testing for multinomial distributions in federated learning in the case where the number of samples n per federated agent is large, and under a bandwidth or privacy constraint. Under certain scaling regimes, the authors characterize the number of samples needed for risk (defined as sum of Type I and Type II error) to vanish asymptotically. The authors provide an excellent description of Le Cam theory, and discuss it’s relation to their work.

**Strengths:**

This paper is very well written. It is both thorough and rigorous, while still being approachable and well-written. Section 4 and 5 specifically are very nicely done and explain a complex idea simply. The results are interesting, and novel to my knowledge.

**Weaknesses:**

The paragraphs after Theorem 1 and 2 respectively could be expanded somewhat. It would be interesting to see some discussion about the theorems in context.

Small Comments

166: the the sample

303: citation missing

**Questions:**

The title says "Optimal Private *and* Communication Constraint Distributed Goodness-of-Fit Testing". However, it seems that you consider "Optimal Private *or* Communication Constraint", as Theorem 1 and 2 consider these constraints separately. Can you comment on this? What would happen if you consider these jointly?

**Limitations:**

I think this paper is thorough, and details the exact theoretical setting where the proven results apply.

---

> ### Author Rebuttal · Authors · 2024-08-07
>
> We sincerely appreciate the time and effort spent on evaluating our paper, the positive feedback and the insightful question. Below, we address the  suggestions and questions raised by the Reviewer.
>
> *"The paragraphs after Theorem 1 and 2 respectively could be expanded somewhat. It would be interesting to see some discussion about the theorems in context."*
>
> We appreciate your suggestion to expand the discussion following Theorems 1 and 2. We agree that there are interesting phenomena that are not remarked on, or could be discussed further. In the revised version, we have expanded the discussion surrounding Theorem 1 and 2 to include the following:
>
> *After Theorem 1:* We have extended our discussion of the rate derived in the theorem when compared to the small $n$ rate. We now elaborate on the phase transition observed in our results, which is not observed in the $n = 1$ regime. We have also added a detailed discussion comparing our results to those of Acharya et al. [1] in the estimation setting. Specifically, we highlight the similar communication super-efficiency phenomenon observed in their work for distributed estimation of discrete distributions under bandwidth constraints. We explain how their results show a transition in the estimation risk depending on the relationship between the number of local observations $n$, the dimension $d$, and the communication budget $b$. Such a super-efficiency is also observed when comparing [6] and [7], but for continuous densities. See also our reply to Reviewer (Nun8).
>
> *After Theorem 2:* Similarly to the result of Theorem 1, the large $n$ problem undergoes phase transitions that are not observed in the $n=1$ version under differential privacy constraints. Even though there is no direct privacy equivalent to the communication super-efficiency phenomenon in Theorem 1, these phase transitions do mean that the problem undergoes significant changes in terms of its dynamics in the large $n$ regime. We discuss the meaning of these phase transitions in more detail in the revised version of the article.
>
> "*The title says "Optimal Private and Communication Constraint Distributed Goodness-of-Fit Testing". However, it seems that you consider "Optimal Private or Communication Constraint", as Theorem 1 and 2 consider these constraints separately. Can you comment on this? What would happen if you consider these jointly?*"
>
> Thank you for this very interesting question. Indeed, Theorem 1 and 2 consider the constraints separately. The reason for this, is that for the Gaussian model for which leverage asymptotic equivalence, only results under both constraints separately have been established. Although we did not explicitly state this in the original version of the manuscript, the general theory we develop in Section A.2 could also be applied to settings in which bandwidth and differential privacy hold simultaneously. We have extended the results of Section A.2 to explicitly include protocols where both constraints hold at the same time. In general, we believe the setting were both constraints hold simultaneously could lead to very interesting questions; for example whether one one of the constraints necessarily more stringent. We know of [8], who consider  a setting with both constraints at the same time, for discrete distributions specifically. However, only the one observation per server setting ($n=1$) is considered in [8], which makes it impossible to leverage their results to the Gaussian model. Regardless, we have incorporated this discussion in the revised version of manuscript.

---

> > ### Comment · Reviewer_aGfz · 2024-08-07
> > **Reply to Rebuttal**
> >
> > Thank you for your response. I will maintain my rating.

---

> > > ### Author Response · Authors · 2024-08-12
> > >
> > > We would like to thank the the Reviewer (aGfz) again for the time and effort spent on evaluating our work and for their response to our rebuttal.

---

### Official Review · Reviewer_Nun8 · 2024-07-17

**Soundness:** 3
**Presentation:** 2
**Contribution:** 2
**Rating:** 4
**Confidence:** 4

**Summary:**

The paper addresses distributed goodness-of-fit testing problems under user-level communication and local differential privacy (DP) constraints. In this scenario, each of the m users receives n samples, and a central server aims to test whether the underlying discrete distribution is uniform. This classical problem has been extensively studied in similar settings, such as when n = 1 or when the task is to estimate the underlying distribution.

The main contribution of this paper is a tight characterization of the separation rates in the large-sample regime (where n is sufficiently large). The primary technical tool employed in the proof is Le Cam's statistical equivalence. By leveraging the equivalence between the Gaussian location model (GLM) and the multinomial model in large local sample regimes, the problem is reduced to the GLM, which has been addressed in previous works.

While Le Cam's statistical equivalence is indeed a powerful and elegant method for establishing minimax rates, I find the novelty and contribution of this work to be somewhat limited. The main technical tools are well-established, and the primary theorem applies only to a restricted parameter regime. Additionally, the discussion of related prior works could be more comprehensive. Lastly, the presentation of the paper could be significantly improved.

============== Post rebuttal =========
I have updated my score accordingly, given that the authors claim that the techniques used in this work can be extended to prove lower bounds under interactive models.

**Strengths:**

The main technical tool, the notion of statistical equivalence, seems to be a suitable and powerful method for addressing this class of problems, as it allows for the reduction of one problem to another.

**Weaknesses:**

1. **Limited Contributions**: The main technical contribution of this paper appears to be limited, as the statistical distance between multinomial models and Gaussian location models is derived from prior works. Additionally, distributed testing for GLM under local DP/communication constraints is also well-established. The main theorems only establish the separation rates in a limited regime (i.e., large \(n\)). The proposed reduction, and consequently the statistical equivalence, apply only to the non-interactive setting, and it is well-known that establishing interactive lower bounds is significantly more challenging.

2. **Insufficient Discussion of Prior Works**: There are several relevant works that need further discussion. For instance, the "communication super-efficiency" effect, which has appeared in the "estimation" version of the problem in [1], is not discussed in the current draft. Although the paper is included in the references, I could not find the corresponding citation in the main text.

3. **Presentation**: The organization and presentation of the paper can be significantly improved. For example, the symbols \(\mathcal{Q}\) and \(\mathcal{P}\) are sometimes used to refer to multinomial and Gaussian models (e.g., in the proof of Theorem 1) without being explicitly stated, and at other times they refer to two general statistical models (e.g., in Section A.2), which is confusing. It would be helpful to explicitly specify these notations. There are also many minor typos and unclear notations that reduce readability. Some examples include:
   - Line 286: "identified" should be "identical"?
   - Line 303: unclear reference []
   - Line 642: missing reference
   - Line 673: missing reference


[1] Archaya et. al., "Distributed estimation with multiple samples per user: Sharp rates and phase transition".

**Questions:**

N/A

---

> ### Author Rebuttal · Authors · 2024-08-07
>
> We express our sincere thanks to the Reviewer for taking the time and effort to thoroughly review, the insightful comments and constructive feedback on our paper. The Reviewer identifies areas for improvement, which we will address point-by-point below.
>
> ### *Limited contribution:*
>
> "*The main technical contribution of this paper appears to be limited, as the statistical distance between multinomial models and Gaussian location models is derived from prior works. Additionally, distributed testing for GLM under local DP/communication constraints is also well-established. The main theorems only establish the separation rates in a limited regime (i.e., large (n)).*"
>
> Federated goodness-of-fit testing for discrete distributions under bandwidth and differential privacy constraints is a notoriously hard problem to study, which has been solved for the case of $n=1$ in a series of articles by Acharya et. al. We believe that the $n \geq 1$ problem is very interesting, as it encompasses a very important class of models in a setting where bandwidth and differential privacy constraints are very natural.
>
> Currently, the tools available in the literature seem insufficient to tackle the problem for all regimes (i.e. small $n$, large $n$ and everything in between). The tools developed by Acharya et. al. are, roughly speaking, combinatorial in nature, which more or less constrains its application to the $n=1$ case. The tools developed in the Gaussian setting recover the full regime, but do not apply outside the Gaussian setting. This is unlike federated estimation setting, which has been solved for "all regimes" and where boardly applicable / general theory for deriving upper- and lower-bounds exist, such as those developed in [3].
> Because we believe the problem to be both very difficult to solve but also highly important. Therefore, we are of the opinion that although our main theorems only establish the minimax separation rates in a limited regime, our paper still provides an important contribution.
>
> "*The proposed reduction, and consequently the statistical equivalence, apply only to the non-interactive setting, and it is well-known that establishing interactive lower bounds is significantly more challenging.*"
>
> Thank you for raising this point. For the problem of goodness-of-fit testing, it is known that there is no benefit to having interactive protocols, when shared randomness is available. Shared randomness, which can be seen as a subset of sequentially interactive protocols, is considered in our work. It is known that shared randomness protocols are sufficient to obtain optimal rates in the problems that we consider in our article, in the sense that e.g. general sequential protocols do not improve upon it performance wise, see for example [2].
>
> That said, we agree that for our general theory, as developed in Section A, interactive protocols are certainly interesting to consider; there known settings where e.g. sequentially interactive protocols strictly improve over non-interactive protocols, such as adaptation (see e.g. [5]). We have been able to extend the theory here to the sequential and blackboard protocols such as considered in [4]. In the revised version of our article, we also discuss interactive protocols more generally, including the works of [2,3,4,5].
>
> ### *Insufficient discussion of prior works:*
>
> We thank the Reviewer for the suggestion of discussing [1] and other related work more expansively. Beyond our introduction, we have used some of the space provided by the additional page of the revised version to provide a more detailed comparison of our results to [1,2,6,7]. Below, we outline the extended discussion of these relevant results.
>
> We cited [1] in our introduction, and we agree that their work is directly relevant to our results. They study what can be seen as the (bandwidth constrained) `estimation version' of the testing problem considered by us. Their results indeed describe a similar communication super-efficiency phenomenon as we find in the testing setting: the small $n$ exponential benefit of additional communication budget is lost as the number of observations locally become large comparatively to $d$. We contrast our main theorems with those of [1] in Section 3: "Minimax rates in the large sample regime" in the revised version of our manuscript.
>
> Such a communication super-efficiency phenomenon can also be observed in continuous data, when contrasting the work of [6] with [7], who study density estimation in a distributed setting under bandwidth constraints. We have included a discussion of these results in our revised version also.
>
> ### *The Presentation:*
> Thank you for your suggestions on how to further improve the presentation of the paper. Indeed, the statements / machinery of Section A of the appendix apply to general models (denoted by $\mathcal{P}$ and $\mathcal{Q}$), whereas in the main text of the article, $\mathcal{P}$ and $\mathcal{Q}$ specifically refer to the Gaussian location model and the multinomial model, respectively. To clarify this, we open Section A in our revised manuscript with the sentence: *"In this section, we present results and theoretical developments that apply to general models denoted by $\mathcal{P}$ and $\mathcal{Q}$. While the main text specifically focuses on the Gaussian location model for $\mathcal{P}$ and the multinomial model for $\mathcal{Q}$, the machinery developed here is applicable to general statistical models."* By doing so, we hope to provide a clear distinction between the general models discussed in Section A and the specific models explored in the main text. If you believe distinct notation, e.g. $\mathcal{P}^{gauss}$ and $\mathcal{Q}^{mult}$ is more helpful, we are also open such a change.
>
> We thank you also for pointing out the typos and missing references. On top of fixing the ones you pointed out, we have conducted a very careful proofread, making sure the revised manuscript does not contain such mistakes.

---

> > ### Comment · Reviewer_Nun8 · 2024-08-12
> > **Reply to the rebuttal**
> >
> > Thank you for your response. I agree that the regime where $n \geq 1 $ is indeed interesting. However, my primary concern remains with the technical novelty of the work. The statistical distance between multinomial models and GLMs has already been established, and distributed testing for GLMs under local DP is known too. As such, the main technical contribution appears to be bridging these two results. While I acknowledge that this involves some non-trivial extension to distributed settings, I still find the overall contribution somewhat limited.
> >
> > Regarding the interactive protocols, I also agree that, as in many other statistical tasks, interaction may not necessarily reduce the error. However, I believe there is currently no established lower bound for the $ n \geq 1 $ regime with sequential or blackboard interaction. In my view, developing such an interactive lower bound would significantly strengthen the work. Nevertheless, based on the current draft and the authors' response, it's unclear whether the framework developed in this paper can be extended to address this scenario.
> >
> > I appreciate the authors' attention to the references and the presentation issues. Please do include a discussion of [1] explaining why their techniques do not apply to the testing problem.

---

> > > ### Author Response · Authors · 2024-08-12
> > >
> > > We thank Reviewer (Nun8) for their response. We are happy to hear that they agree the $n\geq 1$ setting is interesting and that the extension is non-trivial.
> > >
> > > We also find that there seems to be no known lower bound for blackboard or sequential protocols when $n > 1$ in the case of testing. In [2], the authors show that for $n=1$, there is no benefit to a sequential setup when compared to a shared randomness setup for uniformity testing with discrete data. For the Gaussian location model, we note that the proof of the shared randomness lower bounds of [11] can be extended to obtain the same (rate) results for sequential setups as well. Our (extended) machinery of the revised version then explicitly implies that there is no benefit of a sequential setups (outside of shared randomness) for uniformity testing with discrete data in the regime where $md \log (d) / \sqrt{n} = o(1)$. We will add details of this particular extension to the revised version of our paper.
> > >
> > > Admittedly, this does not mean that we can conclude anything concerning the benefit of a sequential setup in the regime(s) where $n > 1$ and $md \log (d) / \sqrt{n} \gtrsim 1$. Also, we note that for blackboard protocols, much less is known and we can indeed not exclude to benefit of a blackboard protocol. Proving lower bounds in the testing setting for blackboard protocols specifically is an interesting but difficult problem with many open questions in the literature.
> > >
> > > We will also include a discussion on why the technique of [1] does not yield an optimal lower bound in the testing setting in our revised version. We agree that including such a discussion is important to highlight the contribution of our work. We briefly sketch the reasons that this (and certain other) estimation techniques do not extend to the testing setting below.
> > >
> > > Let us start with describing a similarity: for both the estimation and the testing problem, lower bounds are typically proven by bounding a divergence measure between probability distributions, such as the chi-square divergence, mutual information or total variation [1,3,7,11,12,13,18,19].
> > >
> > > For estimation problems such as the one considered in [1], or those of the examples considered in [3], it suffices to essentially "tensorize" the divergence, which, loosely speaking, breaks the problem into the ``sum of the local divergences''. This is essentially the role of Theorem C.2 in [1] (see also Theorem 1 and 2 in [3]), which bounds the total variation between elements of a perturbed family of probability distributions by the sum of the local conditional "scores", local conditional variances of the transcript densities or the local mutual information, see (16), (17) and (18) in [1]'s supplement. The loss due to a bandwidth constraints (in [1]) or privacy constraint (in [3]) is then captured by data processing arguments. For estimation, such tensorization bounds turn out to give tight lower bounds. We note that a lot more goes into the proof of [1] (e.g. Poissonisation, sub-Gaussian concentration), but the principle difference with estimation and testing is this kind of tensorization step. Similar tensorization arguments can also be found in other estimation problems such as [4,7], for the Fisher information and mutual information respectively.
> > >
> > > Such a "tensorization approach" does not yield tight bounds in testing problems. Using mutual information, [19] tries this tensorization approach for the testing problem, but they only recover the optimal testing rates when each server communicates only one bit ($b=1$). Another "estimation" approach tried for a goodness-of-fit testing problem can be found [18], which similarly obtain a lower bound for testing that is only tight for $b=1$, through a direct Taylor expansion of the likelihood (which can also be seen as tensorizing a divergence). The authors of [18] provide a detailed discussion of the shortcomings of the latter approach in Section 4 of their paper.
> > >
> > > To obtain tight lower bounds for the testing problem, the papers that successfully do so for $b > 1$ and privacy constraints (i.e. [11,12,13]) use techniques that differ greatly from the techniques employed in [1,3]. In [13], the authors use a combinatorial expansion of the likelihood that works specifically for $n=1$ in the multinomial model, which does not generalize to large numbers of observations. [11,12] circumvent the latter issue in the Gaussian setting by employing a Brascamp-Lieb inequality, an inequality from functional analysis. This inequality explicitly uses the Gaussianity of the log-likelihood explicitly.
> > >
> > > We hope that above additions to our work are satisfactory and we would like to thank you again for your consideration of our work.
> > >
> > > Additional references:
> > >
> > > [18] Acharya et. al., "Distributed signal detection under communication constraints"
> > >
> > > [19] Szabo et. al., "Optimal distributed composite testing in high-dimensional Gaussian models with 1-bit communication"

---

### Author Rebuttal · Authors · 2024-08-07

First of all, we would like to thank the Reviewers for carefully reading our paper and their interest in our work. We are happy to hear that the majority of the Reviewers found our paper "very well written" (aGfz), "well-written, well-organized" (3Xr9) and the theory derived "rigorous and thorough" (4qfs), "mathematically rigorous" (3Xr9), "thorough and rigorous, while still being approachable" (aGfz). We are also delighted to hear that our paper "addresses key challenges in distributed settings" (4qfs), and that "the results are interesting" (aGfz) and novel (q4fs, aGfz).

The Reviewers have also raised a few concerns and provided several suggestions which we have addressed point-by-point in the individual rebuttals. Here we collect the main changes in the manuscript.

* We have extended the discussion of our results, following the suggestions of each of the Reviewers (see the comments below for details). We have also followed suggestions on further improving the presentation and clarity.

* We extended the general theoretical machinery developed in Section A of the supplement, showing that these tools apply to the other federated settings; such as sequential protocols, blackboard protocols and settings where a bandwidth constraint and a differential privacy constraint are jointly imposed.

* Following suggestions of the Reviewers, we have added references to existing literature and have provided additional discussion of existing articles. Besides exploring interesting contrasts between existing work with ours, this also provides additional context for our work.

* The reviewers identified some typos and missing references. In addition to addressing these issues, we conducted a thorough proofreading to ensure that the revised manuscript is free of such errors.

We would like to thank all the Reviewers for their consideration.

---

References discussed in the individual rebuttals:

[1] Acharya et. al., "Distributed estimation with multiple samples per user: Sharp rates and phase transition".

[2] Acharya et. al., "Interactive Inference under Information Constraints".

[3] Acharya et. al., "Unified Lower Bounds for Interactive High-dimensional Estimation under Information Constraints"

[4] Barnes et. al., "Fisher Information for Distributed Estimation under a Blackboard Communication Protocol"

[5] Cai et. al., "Distributed adaptive gaussian mean estimation with unknown variance: interactive protocol helps adaptation"

[6] Han et. al., "Distributed Statistical Estimation of High-Dimensional and Nonparametric Distributions"

[7] Szabo and Zaman, "Distributed nonparametric estimation under communication
constraints."

[8] Surin Ahn et. al., "Estimating Sparse Distributions Under Joint Communication and Privacy Constraints"

[9] Canonne et. al., "Private identity testing for high-dimensional distributions."

[10] Narayanan, "Private high-dimensional hypothesis testing"

[11] Szabo et. al., "Optimal high-dimensional and nonparametric distributed testing under communication constraints"

[12] Cai et. al., "Federated Nonparametric Hypothesis Testing with Differential Privacy Constraints: Optimal Rates and Adaptive Tests"

[13] Acharya et. al., "Lower Bounds from Chi-Square Contraction"

[14] Carter, "Deficiency distance between multinomial and multivariate normal experiments"

[15] Cai et. al., "The cost of privacy: Optimal rates of convergence for parameter estimation with differential privacy"

[16] Chhor and Sentenac, "Robust Estimation of Discrete Distributions under Local Differential Privacy"

[17] Duchi et. al., "Local Privacy and Statistical Minimax Rates"

---

### Decision · Program_Chairs · 2024-09-25

**Decision:**

Accept (poster)

**Comment:**

This paper studies distributed goodness-of-fit testing for discrete distribution under bandwidth and differential privacy constraints. The authors consider the important extension from the $n=1$ case studied in detail by Archaya and co-authors to the interesting regime of $n\ge 1$ where some new techniques need to be employed. The authors derive matching minimax upper- and lower-bounds for the goodness-of-fit testing. Generally, reviewers acknowledge that the paper is interesting and makes some advancements, though one questioned the amount of novelty going from the $n=1$ case to the $n\ge 1$ case. The AC believes this paper crosses the bar and should be of interest to the statistical learning community at NeurIPS.